# Consistency-diversity-realism Pareto fronts of conditional image generative models

## Abstract

Building *world models* that accurately and comprehensively represent the real world is a holy grail for image generative models as it would enable their use as world simulators. For conditional image generative models to be successful world models, they should not only excel at image quality and prompt-image consistency but also ensure high representation diversity. However, current research in generative models mostly focuses on creative applications that are predominantly concerned with human preferences of image quality and aesthetics. We note that generative models have inference time mechanisms – or *knobs* – that allow the control of generation consistency, quality, and diversity. In this paper, we use state-of-the-art text-to-image and image&text-to-image models and their knobs to draw consistency-diversity-realism Pareto fronts that provide a holistic view on consistency-diversity-realism multi-objective. Our experiments suggest that realism and consistency can both be improved simultaneously; however there exists a clear tradeoff between realism/consistency and diversity. By looking at Pareto optimal points, we note that earlier models are better at representation diversity and worse in consistency-realism, and more recent models excel in consistency-realism while decreasing significantly the representation diversity. By computing Pareto fronts on a geodiverse dataset, we find that the first version of latent diffusion models tends to perform better than more recent models in all axes of evaluation, and there exist pronounced consistency-diversity-realism disparities between geographical regions. Overall, our analysis clearly shows that there is *no best model* and the choice of model should be determined by the downstream application. With this analysis, we invite the research community to consider Pareto fronts as an analytical tool to measure progress towards world models.

## 1 Introduction

Progress in foundational vision-based machine learning models has heavily relied on large-scale Internet-crawled datasets of real images (Schuhmann et al., 2022). Yet, with the acceleration of research on generative models and the unprecedented photorealistic quality achieved by recent text-to-image generative models (Podell et al., 2023; Esser et al., 2024; Ramesh et al., 2022; Saharia et al., 2022), researchers have started exploring their potential as *world models* that generate images to train downstream representation learning models (Astolfi et al., 2023; Hemmat et al., 2023; Tian et al., 2024).

World models aim to represent the real world as accurately and comprehensively as possible. Therefore, visual world models should not only be able to yield *high quality* image generations, but also generate content that is representative of the *diversity* of the world, while ensuring *prompt consistency*. However, state-of-the-art conditional image generative models have mostly been optimized for human preference, and thus, a single high-quality and consistent sample fulfills the current optimization criteria. This vastly disregards representation diversity (Hall et al., 2024; Sehwag et al., 2022; Zameshina et al., 2023; Corso et al., 2024; Hemmat et al., 2023; Sadat et al., 2024), and questions the potential of state-of-the-art conditional image generative models to operate as effective world models. Optimizing for human preferences only partially fulfills the multi-objective optimization required to leverage conditional generative models as world models.

At the same time, state-of-the-art conditional image generative models have built-in inference time mechanisms, hereinafter referred to as *knobs*, to control the realism (also referred to as quality or fidelity),

consistency, and diversity dimensions of the generation process. For example, it has been shown that the guidance scale in classifier free guidance of diffusion models (Ho & Salimans, 2021), trades image fidelity for diversity (Saharia et al., 2022; Corso et al., 2024). Similarly, post-hoc filtering (Karthik et al., 2023) is used to improve consistency. Although recent works have carried out extensive evaluations of image generative models (Ku et al., 2024; Lee et al., 2024), these evaluations have been primarily designed from the perspective of creative applications. To the best of our knowledge, a comprehensive and systematic analysis of the effect of the knobs controlling the different performance dimensions of conditional image generative models has not yet been carried out.

In this paper, we benchmark conditional image generative models in terms of the world models' multi-objective. In particular, we perform an optimization over both knobs and state-of-the-art models with the goal of capturing the consistency-diversity, realism-diversity, and consistency-realism Pareto fronts that are currently reachable. In our analysis, we include both text-to-image (T2I) models and image&text-to-image (I-T2I) models. For T2I, we consider several version of latent diffusion models (LDM), namely $LDM_{1.5}$ and $LDM_{2.1}$ (Rombach et al., 2022), as well as $LDM_{XL}$ (Podell et al., 2023), whereas for I-T2I, we consider a retrieval-augmented diffusion model (RDM) (Blattmann et al., 2022) and $LDM_{2.1\text{-}UnCLIP}$ (Ramesh et al., 2022), in addition to a neural image compression model, PerCo (Careil et al., 2024). We perform our T2I and I-T2I models analysis using the ubiquitous MSCOCO (Lin et al., 2014) validation dataset and we extend our evaluation of T2I models to the GeoDE dataset (Ramaswamy et al., 2024), composed of images from 6 world regions, to characterize the progress of these models from a geographic representation perspective. To quantify the multi-objective, we use inter-sample similarity and recall (Kynkäänniemi et al., 2019) to measure representation diversity; image reconstruction quality and precision (Kynkäänniemi et al., 2019) to quantify realism; and the Davidsonian scene graph score (Cho et al., 2024)) to assess prompt-generation consistency.

By drawing the Pareto fronts, we observe that progress in conditional image generative models has been driven by improvements in image realism and/or prompt-generation consistency, and that these improvements result in models sacrificing representation diversity. In the T2I setup, our analysis suggests that more recent models should be used when the downstream task requires samples with high realism – $LDM_{XL\text{-}Turbo}$– and consistency – $LDM_{XL}$–. However, older models – $LDM_{1.5}$ and $LDM_{2.1}$– are preferable for tasks that require good representation diversity. For I-T2I models, we observe that compression models – *e.g.*, PerCo– should be prioritized when working on downstream applications that require high realism and consistency. However, when the downstream application requires high representation diversity, RDM and $LDM_{2.1\text{-}UnCLIP}$ are preferable. Interestingly, on GeoDE we observe that the oldest model, $LDM_{1.5}$, outperforms the most recent ones, and consistently appears in the Pareto fronts of all regions considered. Moreover, the advances in T2I models reduce region-wise disparities in terms of consistency and increase the disparities in terms of image realism, while sacrificing diversity across all regions. Finally, by looking at the knobs, we observe that guidance and post-hoc filtering have the highest effect on the consistency-diversity and realism-diversity tradeoffs, increasing both realism and consistency at the expense of representation diversity. We believe that the proposed evaluation framework and the findings that arise from it will enable faster progress towards enabling the use of conditional image generative models as world models, and we hope it will encourage the research community to work on models that present softer consistency-diversity-realism tradeoffs.

## 2 Methodology of the analysis

In this section, we introduce the notation adopted throughout the rest of the paper, present the metrics we use to evaluate conditional image generative models, and describe existing knobs that control the consistency-diversity-realism multi-objective. We refer to Appendix A for a comparison with related work.

**Notation.** Let us consider the following conditional image generation framework. An image generative model, $g_\theta$, parameterized by a set of learnable parameters, $\theta$, generates an image, $Y$, from a noise sample $Z \sim \mathcal{N}(0, \mathbf{I})$ and a conditioning prompt encoded by a vector, $\mathbf{p} \in \mathbb{R}^d$: $Y = g_\theta(Z, \mathbf{p})$. In state-of-the-art conditional image generative models, $\mathbf{p}$ encodes either text, an image, or a combination of both. In practice, images are generated in batches of $N$ elements, $\mathbf{Y} \in \mathbb{R}^{N \times H \times W \times 3}$, conditioned on the *same* vector $\mathbf{p}$ and a tensor $\mathbf{Z} \in \mathbb{R}^{N \times H \times W \times 3}$ representing $N$ random noise samples:

$$\mathbf{Y} = g_\theta(\mathbf{Z}, \mathbf{p}). \tag{1}$$

## 2.1 Evaluating conditional image generation

We evaluate conditional image generation in terms of prompt-sample consistency, sample diversity and realism (also referred to as quality or fidelity in the literature). We consider two complementary ways of quantifying the performance of conditional image generative models: conditional and marginal. On the one hand, *conditional metrics* are prompt-specific scores computed on the set of image generations resulting from a prompt. An overall score may be obtained by averaging out all prompt-specific scores. On the other hand, *marginal metrics* are overall scores computed on the generations resulting from *all* the prompts directly. In practice, marginal metrics compare a set of generated images to a reference dataset while ignoring the prompts used to obtain the sets. In the reminder of this subsection, we define consistency – that is always conditional –, conditional and marginal diversity, as well as conditional and marginal realism.

**Consistency, $\mathcal{C}$.** Prompt-generation consistency is measured either with distance or similarity-based scores – *e.g.*, CLIPScore (Hessel et al., 2021), LPIPS score (Zhang et al., 2018) and DreamSim score (Fu et al., 2023) – or with visual question answering (VQA) approaches – *e.g.*, TIFA (Hu et al., 2023), VQAScore (Lin et al., 2024), and DSG (Cho et al., 2024) metrics –. In our analysis, we opt to use VQA approaches as they are reported to be more calibrated and interpretable than the distance and similarity-based scores (Cho et al., 2024). Concretely, we measure the prompt-generation consistency with Davidsonian Scene Graph (DSG) score. DSG relies on binary questions $\mathbf{Q}$ and answers $\mathbf{A}$ generated via LLMs based on the image prompt $\mathbf{p}$. A vision-language model finetuned for the VQA task, VQA($\cdot$), is fed with an image, $\mathbf{Y}$, and the set of questions of its respective prompt. The VQA model predict answers, which are compared to the ground-truth to get a score. The resulting formulation for per-prompt consistency, $\mathcal{C}^p$, reads as:

$$\mathcal{C}^p = \frac{1}{N} \sum_{j=1}^{N} \frac{1}{Q_j} \sum_{i=1}^{Q_j} \mathbb{1}\Big(\text{VQA}(\mathbf{Y}_j, \mathbf{Q}_i), \mathbf{A}_i\Big), \tag{2}$$

where $N$ represents the number of images generated per conditioning prompt, $Q_j$ represents the number of question per j-th image, and $\mathbb{1}$ represents the indicator function. The consistency over a set of prompts may be aggregated into a global consistency score, $\mathcal{C}$, by averaging all the conditioning-wise DSG scores, $\mathcal{C}^p$.

**Conditional diversity, $\mathcal{D}_C$.** We measure per-prompt conditional diversity as follows:

$$\mathcal{D}_C^p = \frac{1}{N^2 - N} \sum_{j \neq i} \mathcal{S}(f_\phi(\mathbf{Y}_j), f_\phi(\mathbf{Y}_i)), \tag{3}$$

where $\mathcal{S}$ is a similarity or distance function, and $f_\phi$ is an image feature extractor. In our analysis, we use cosine similarity and the DreamSim (Fu et al., 2023) feature extractor. DreamSim leverages an ensemble of modern vision encoders, including DINO (Caron et al., 2021) and two independently trained CLIP encoders, and is reported to correlate well with human perception. The conditional diversity over a set of prompts may be aggregated into a global score, $\mathcal{D}_C$, by averaging all the conditioning-wise scores, $\mathcal{D}_C^p$.

**Conditional realism, $\mathcal{R}_C$.** We measure per-prompt conditional realism as follows:

$$\mathcal{R}_C^p = \frac{1}{N} \sum_{j=1}^{N} \max_i(\mathcal{S}(f_\phi(\mathbf{X}_i), f_\phi(\mathbf{Y}_j))), \quad i \in \{1, \ldots, N'\}, \tag{4}$$

where $\mathbf{X} \in \mathbb{R}^{N' \times H \times W \times 3}$ represents a tensor of $N'$ real images. Note that both $\mathbf{X}$ and $\mathbf{Y}$ represent generations and real images of the same prompt $\mathbf{p}$, respectively. Similarly to conditional diversity, we implement $\mathcal{S}$ as cosine similarity and use DreamSim as feature extractor. The conditional realism over a set of prompts may be aggregated into a global score, $\mathcal{R}_C$, by averaging all the conditioning-wise scores $\mathcal{R}_C^p$.

**Marginal diversity, $\mathcal{D}_M$.** Commonly used metrics of marginal diversity, such as *recall* (Sajjadi et al., 2018; Kynkäänniemi et al., 2019) or *coverage* (Naeem et al., 2020), compare real and generated image distributions by leveraging a reference dataset of real images to ground the notion of diversity. Marginal diversity may also be measured with metrics which do not rely on a reference dataset, such as the Vendi Score (Friedman & Dieng, 2023). In our analysis, we use recall (Sajjadi et al., 2018; Kynkäänniemi et al., 2019) to compute

marginal diversity given its ubiquitous use in the literature. Recall measures marginal diversity as the probability that a random real image falls within the support of the generated image distribution.

**Marginal realism, $\mathcal{R}_M$.** The most commonly used metric to estimate image realism is the Fréchet Inception Distance (FID) (Heusel et al., 2017). FID relies on a pre-trained image encoder (usually, the Inception-v3 model trained on ImageNet-1k (Szegedy et al., 2015)) that embeds both generated and real images from a reference dataset. The metric estimates the distance between distributions of features of real images and features of generated images, relying on a Gaussian distribution assumption. The FID summarizes image realism and diversity into a single scalar. In our analysis, to disentangle both axes of evaluation, we use precision (Kynkäänniemi et al., 2019; Naeem et al., 2020) as marginal realism metric. Precision measures marginal realism as the probability that a random generated image falls within the support of the real image distribution.

## 2.2 Consistency-diversity-realism knobs

**Guidance scale.** To control the strength of the conditioning, a guidance scale (g-scale) hyper-parameter can be used to bias the sampling of diffusion models like DDPM (Ho et al., 2020), see *e.g.*, classifier (Dhariwal & Nichol, 2021) or classifier-free guidance (CFG) (Ho & Salimans, 2021). More precisely, rewriting Eq. (1) for diffusion models trained with CFG, we obtain:

$$\mathbf{Y} = \lambda g_\theta(\mathbf{Z}, \mathbf{p}) + (1 - \lambda) g_\theta(\mathbf{Z}, \emptyset), \tag{5}$$

where $\lambda$ is the guidance scale, $\emptyset$ is an empty conditioning prompt, and the first and second terms indicate conditional and unconditional samplings, respectively. Importantly, $\lambda$ can be arbitrarily increased ($> 1$) in order to steer the model to generate samples more aligned with the conditioning $\mathbf{p}$.

**Post-hoc filtering.** To improve the generated images, *e.g.* in terms of realism or consistency, or to avoid certain undesirable generations, a set of images generated for the same prompt may be filtered to retain the top-$m$ images based on a predefined criterion, which can be either based on human preferences or automatic metrics. Considering the latter case, a common choice of metric is the CLIPScore, resulting in:

$$\mathbf{Y} = \text{top}\bigg( m, \ \mathcal{S}(\mathbf{p}, f_\phi(\mathbf{Y}_j)) \bigg), \tag{6}$$

where decreasing $m$ ensures higher consistency.

**Retrieval-augmented generation.** Generation can be conditioned on additional information, *e.g.* via nearest-neighbor search in a database given a query image or prompt.

$$\mathbf{Y} = g_\theta(\mathbf{Z}, \oplus_{\mathbf{p}_j \in \mathcal{K} \cup \{\mathbf{p}\}}), \tag{7}$$

where $\oplus$ denotes the aggregation operator and $\mathcal{K}$ is the set of nearest neighbors of $\mathbf{p}$. Existing retrieval-augmented image generative models adopt different aggregation operators. For instance, RDM (Blattmann et al., 2022), KNN-Diffusion (Sheynin et al., 2023), and Re-Imagen (Chen et al., 2022), concatenate the retrieved vectors, and use cross-attention to condition the generative process. Autoregressive models like RA-CM3 (Yasunaga et al., 2023) and CM3Leon (Yu et al., 2023), concatenate the retrieved vectors to the input before performing self-attention. Regardless of the type of aggregation, changing the value of $k$ in retrieval-augmented models can affect the conditional diversity and consistency of the generations.

**Compression rate.** Neural image compression models are generative autoencoder-like models that learn to compress images into low-dimensional representations before reconstructing them. The compression rate, usually expressed in bits-per-pixel (bpp), determines the ability to faithfully reconstruct the original image. Some neural image compression models, such as PerCo (Careil et al., 2024), treat compression as a conditional generative modeling problem, allowing to sample approximate reconstructions given the compressed image code. In such cases, we could expect that by reducing the bitrate, the model might trade consistency/realism for conditional diversity as the compressed image code will carry less information about the original image.

Table 1: Knobs for text-to-image (T2I) and text&image-to-image (I-T2I) models used in our study. For RDM, 1.3M corresponds to the training dataset, while 20M to the retrieval database.

| Model | Dataset size | Knobs | | | |
|---|---|---|---|---|---|
| | | g-scale | top-$m$ filtering | $k$-neighbors | comp. rate |
| *T2I* | | | | | |
| LDM$_{1.5}$ (Rombach et al., 2022) | ∼2B | ✓ | ✓ | ✗ | ✗ |
| LDM$_{2.1}$ (Rombach et al., 2022) | ∼2B | ✓ | ✓ | ✗ | ✗ |
| LDM$_{XL}$ (Podell et al., 2023) | ∼2B | ✓ | ✓ | ✗ | ✗ |
| LDM$_{XL\text{-}Turbo}$ (Sauer et al., 2023) | ∼2B | ✗ | ✓ | ✗ | ✗ |
| *I-T2I* | | | | | |
| PerCo (Careil et al., 2024) | ∼300M | ✓ | ✓ | ✗ | ✓ |
| RDM (Blattmann et al., 2022) | 1.3M + 20M | ✓ | ✓ | ✓ | ✗ |
| LDM$_{2.1\text{-}UnCLIP}$ (Ramesh et al., 2022) | ∼2B | ✓ | ✓ | ✗ | ✗ |

## 2.3 Pareto fronts

We perform an optimization over state-of-the-art models and their knobs with the goal of capturing the consistency-diversity, realism-diversity, and consistency-realism Pareto fronts that are currently reachable, and building understanding on the consistency-diversity-realism multi-objective. More precisely, we quantify consistency, diversity and realism for each pair of (model, knob-value) using the metrics presented in Section 2.1. We then leverage all the resulting measurements to obtain the Pareto fronts that capture the optimal consistency-diversity-realism values achieved by current state-of-the-art conditional image generative models. For visualization ease, we transform the multi-objective into three bi-objectives: consistency-diversity, realism-diversity and consistency-realism.

## 3 Experiments

In this section, we study T2I and I-T2I models and depict the achievable consistency-diversity-realism Pareto fronts by altering the models and their associated knobs. We start by detailing the experimental setups and follow with a detailed discussion of results, covering T2I models in Section 3.1 and I-T2I models in Section 3.2. We then highlight the utility of our approach in a geodiversity analysis in Section 3.3. Finally, we study the impact of using different knobs to control these tradeoffs in Section 3.4.

**Models.** We consider different state-of-the-art conditional image generative models and group them by their conditioning modalities. For T2I models, we consider several versions of LDM: LDM$_{1.5}$, LDM$_{2.1}$ (Rombach et al., 2022), LDM$_{XL}$ (Podell et al., 2023)[1], and LDM$_{XL\text{-}Turbo}$ (Sauer et al., 2023). For I-T2I models, we pick LDM$_{2.1\text{-}UnCLIP}$ (Ramesh et al., 2022), RDM (Blattmann et al., 2022), and the neural image compression model PerCo (Careil et al., 2024), which conditions an LDM with a quantized image representation together with its caption[2]. We summarize the models considered in our analysis and their knobs in Tab. 1.

**Datasets.** We benchmark the models on a popular computer vision dataset, MSCOCO (Lin et al., 2014; Caesar et al., 2018). In particular, we use the validation set from the 2014 split (Lin et al., 2014), which contains 41K images, to compute the marginal metrics, and the 2017 split (Caesar et al., 2018), which contains 5K images, to compute the conditional metrics. This choice is mostly to limit computational costs, as conditional metrics require multiple samples for each conditioning. In addition, we benchmark geographic representation with GeoDE (Ramaswamy et al., 2024), which contains images from everyday objects in countries across six geographic regions. Following Hall et al. (2024), we balance the dataset across 27 objects, yielding 29K images and 162 unique {object} in {region} prompts.

**Implementation details.** We adopt the `Diffusers` library for the LDM models (von Platen et al., 2022) and the official models' repos for RDM and PerCo. We set the number of inference steps to 50

---
[1]For LDM$_{XL}$ we use the base model v1.0 without the refiner
[2]We note that PerCo usually caption the input image with a captioner, while in our case we get the caption from the dataset.

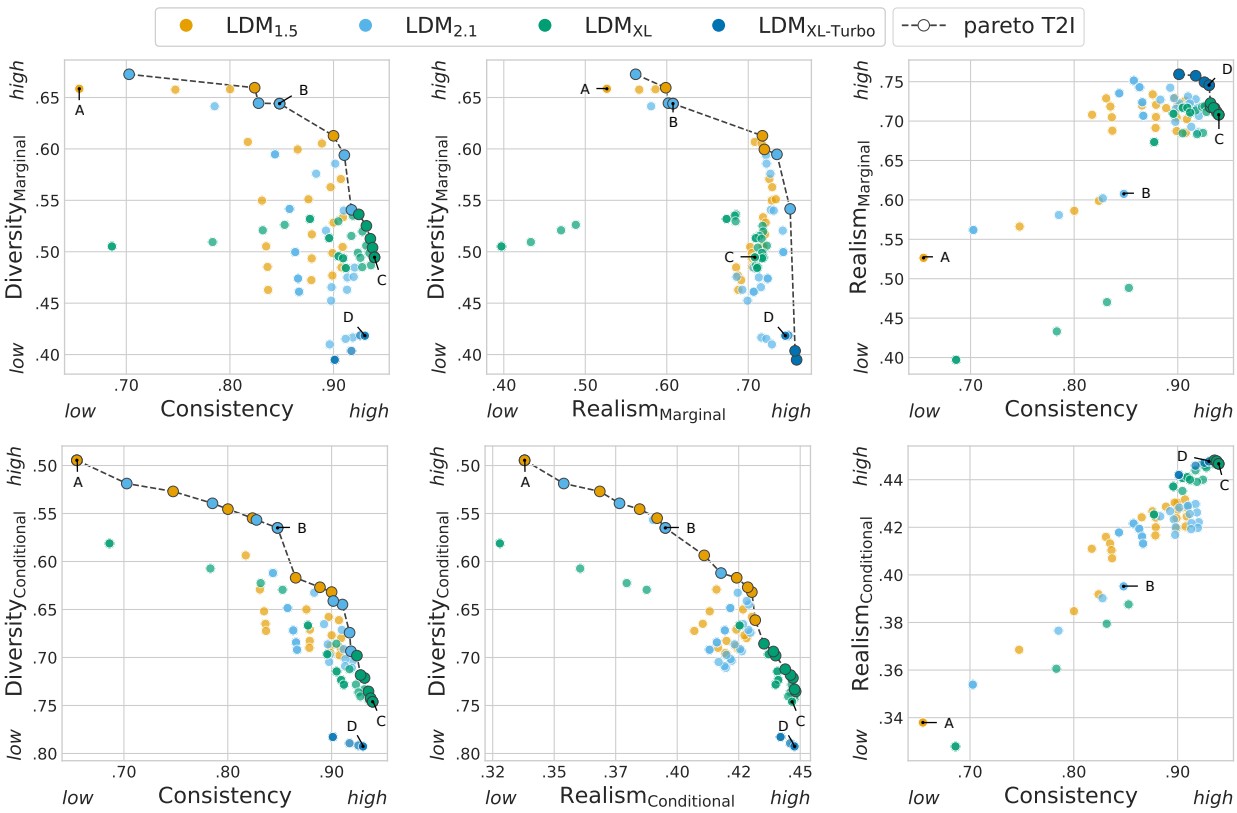

Figure 1: Consistency-diversity (left), realism-diversity (middle) and consistency-realism (right) Pareto fronts for T2I generative models. (top) marginal, (bottom) conditional metrics. Each dot is a configuration of model's knobs. Labeled dots (A-D) are visualized in Fig. 2.

(20 for PerCo as suggested in their paper) using deterministic sampling strategies, DPM++ (Lu et al., 2022) for `Diffusers` models and DDIM (Song et al., 2020) for others. For the conditional metrics on MSCOCO, we sample 10 images per prompt, using the 5,000 image-caption pairs of the 2017 validation split, while for the marginal metrics we sample 1 image per conditioning, using 30,000 randomly selected data points from the validation set of 2014. Note that, as MSCOCO contains multiples captions for each image, we fix the first caption as prompt for generations. For GeoDE, we sample 180 images for each of the `{object} in {region}` prompts for both conditional and marginal metrics. We disaggregate metrics by groups, per Hall et al. (2024), to measure disparities between geographic regions. For metrics based on DreamSim we use the ensemble backbone as recommended from the official repository. For marginal metrics we use improved precision and recall (Kynkäänniemi et al., 2019) with 5 nearest neighbors and Inception-V3 (Szegedy et al., 2015) features, using the implementation of `prdc`. For DSG, we leverage `GPT-3.5-turbo` to generate questions from the prompts, and `InstructBLIP` (Dai et al., 2024) to make the predictions. When performing top-$m$ filtering we use CLIPScore with `CLIP-ViT-H-14-s32B-b79K` from Hugging Face. Moreover, in Appendix C, we report results ablating different marginal and conditional metrics. Finally, we ablate different values for each knob as reported in Appendix B.

### 3.1 Consistency-diversity-realism multi-objective for text-to-image models

In Fig. 1, we depict consistency-diversity, realism-diversity and consistency-realism Pareto fronts for open source T2I generative models. In particular, Fig. 1 (top) depicts marginal realism and diversity metrics while Fig. 1 (bottom) shows their conditional counterparts. Note that consistency is computed in the same way (DSG) in both figures. We now discuss each of the pair-wise metrics Pareto fronts.

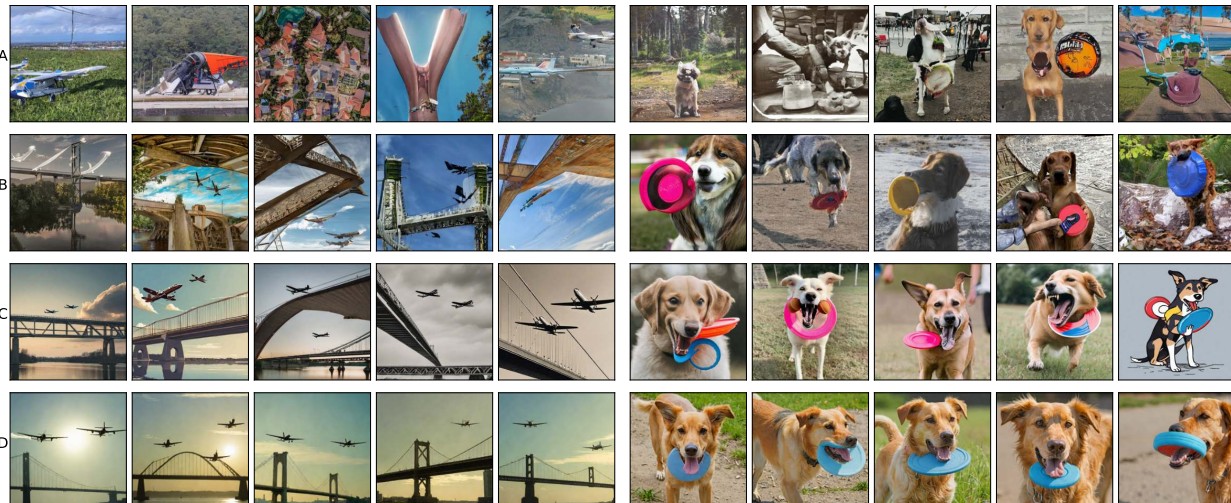

Figure 2: T2I qualitative results on MSCOCO. A-D refer to the models marked in Fig. 1. (left) `Two planes flying in the sky over a bridge.` (right) `There is a dog holding a Frisbee in its mouth.`

**Consistency-diversity.** The Pareto fronts in Fig. 1 (left, top and bottom), are composed of three models: $LDM_{1.5}$, $LDM_{2.1}$ and $LDM_{XL}$. We observe that improvement in diversity, both marginal (Recall) and conditional (DreamSim score), comes at the expense of consistency (DSG). On the one hand, $LDM_{2.1}$ and $LDM_{1.5}$ achieve the best marginal and conditional diversities, respectively. On the other hand, and perhaps unsurprisingly, $LDM_{XL}$ reaches the best consistency ( $\geq 95\%$ of DSG accuracy), while $LDM_{1.5}$ and $LDM_{2.1}$ dominate the middle region of the frontier. Moreover, by comparing these two models, we notice that Pareto optimal hyperparameter configurations of $LDM_{2.1}$ obtain slightly higher consistency scores. In Fig. 2, we validate these observations showcasing samples from $LDM_{1.5}$ (A) at high-diversity/low-consistency, $LDM_{2.1}$ (B) from the middle of the frontier, and $LDM_{XL}$ (C) at high-consistency/low-diversity. Both in the case of the "two planes" and of the "dog", the variance of colors and backgrounds are reduced when visual quality is increased. Other samples are in Appendix C.

**Realism-diversity.** The marginal realism-diversity (Precision-Recall) Pareto front in Fig. 1 (middle, top), is composed of three models: $LDM_{1.5}$, $LDM_{2.1}$ and $LDM_{XL-Turbo}$. In this case, we also observe a tradeoff: higher marginal diversity coincides with lower realism for $LDM_{1.5}$ and $LDM_{2.1}$. $LDM_{XL-Turbo}$ obtains the samples of highest realism. However, we observe that the realism gain compared to $LDM_{2.1}$ is rather small and leads to a steep decrease in sample diversity. We attribute this drop to the adversarial objective used to distill $LDM_{XL-Turbo}$ from $LDM_{XL}$, as also noted by Sauer et al. (2023). Interestingly, $LDM_{XL}$ does not appear on the Pareto front, and it is even quite far away from it. This is probably due to $LDM_{XL}$ (without refiner) generating smooth images lacking of high frequency details (*e.g.*, see the dog in Fig. 2 and (Podell et al., 2023)), and the marginal metrics, which are computed with InceptionV3 features, are sensitive to those frequencies (Geirhos et al., 2018). Instead, by looking at the conditional metrics in Fig. 1 (middle), which are based on DreamSim that extract more sematical features (Fu et al., 2023), we observe that $LDM_{XL}$ belongs to the Pareto front together with $LDM_{1.5}$, $LDM_{2.1}$. In particular, $LDM_{XL}$ achieves the best conditional realism, obtained at the expense of conditional diversity. Here, we remark that $LDM_{XL-Turbo}$ only gets comparable (slightly lower) realism but considerably lower diversity. This difference is evident by looking at C ($LDM_{XL}$) vs. D ($LDM_{XL-Turbo}$) in Fig. 2. When comparing Pareto optimal points of $LDM_{1.5}$ and $LDM_{2.1}$, we note that $LDM_{1.5}$ reaches slightly better conditional realism than $LDM_{2.1}$.

**Consistency-realism.** In Fig. 1 (right, top and bottom) we observe that realism and consistency show relatively strong positive correlation as improvement in one metric oftentimes leads to an improvement in the other metric, with the correlation being stronger for the conditional metrics than for the marginal ones. We observe that the Pareto front is dominated by $LDM_{XL}$ and $LDM_{XL-Turbo}$ model, highlighting how the advancement of T2I generative models have favored consistency-realism over the diversity objective. Indeed, we can also notice that in the distribution of non-Pareto-optimal points, $LDM_{2.1}$ seems better than $LDM_{1.5}$, matching the historical development of these models.

---

**Key insights**

- Progress in T2I models has been driven by improvements in realism and/or consistency. State-of-the art T2I models improve consistency and/or realism by sacrificing representation diversity. Yet, improvements in realism are correlated with improvements in consistency.
- More recent models should be used when the downstream task requires samples with high realism – $LDM_{XL-Turbo}-$ and consistency – $LDM_{XL}-$. However, older models – $LDM_{1.5}$ and $LDM_{2.1}-$ are preferable for tasks that require good representation diversity.
- Both marginal and conditional metrics display correlated Pareto fronts.

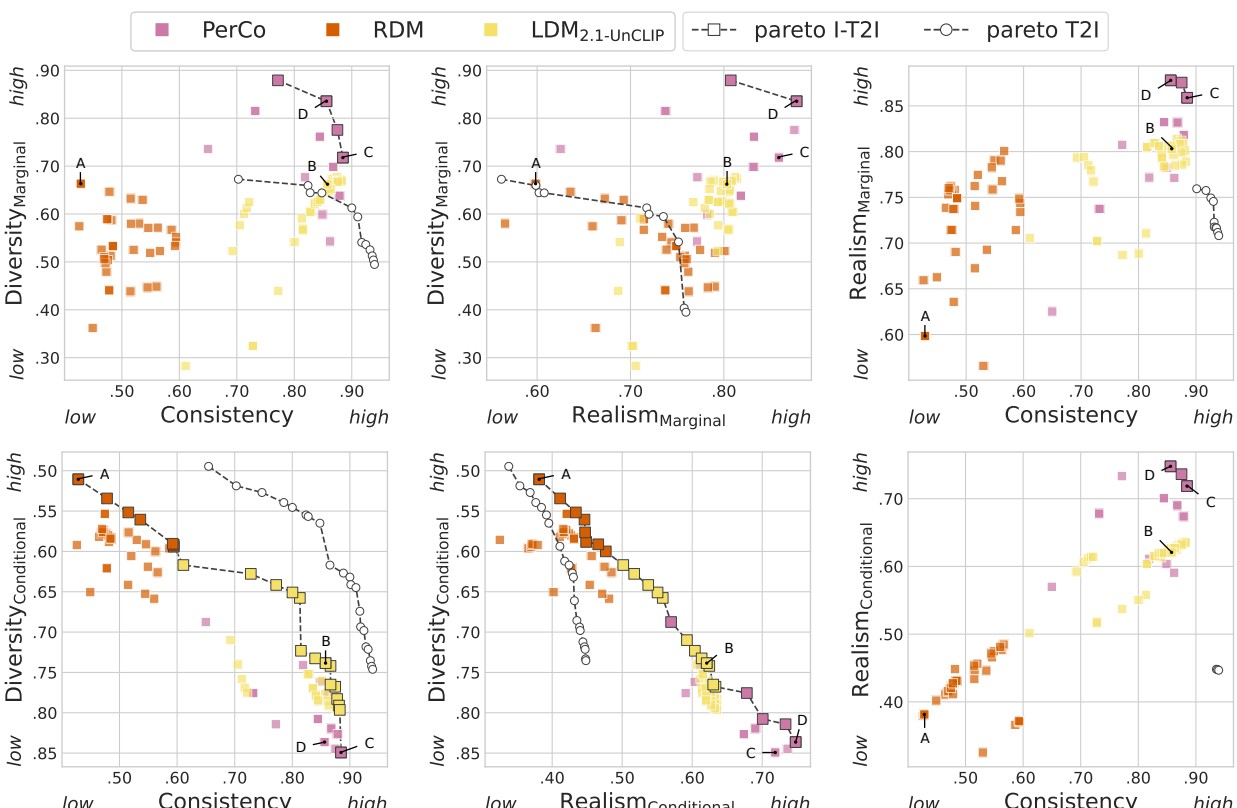

Figure 3: Consistency-diversity (left), realism-diversity (middle) and consistency-realism (right) Pareto fronts for I2I and I-T2I generative models. (top) marginal, (bottom) conditional metrics. Each dot is a configuration of model's knobs. Labeled dots are visualized in Fig. 4

## 3.2 Pareto fronts of image&text-to-image models

**Consistency-diversity.** The marginal consistency-diversity Pareto front in Fig. 3 (left, top) does not show a clear tradeoff, as it is only composed by PerCo neural compression models achieving both high consistency and diversity. On the contrary, for the conditional metrics (left, bottom), the tradeoff is clearly noticeable. The Pareto front is composed of three models: RDM, $LDM_{2.1-UnCLIP}$, and PerCo, RDM reaching the best conditional diversity, $LDM_{2.1-UnCLIP}$ populating a large portion – from mid to high consistency – of the tradeoff, and PerCo achieving the highest consistency, but only for a small margin. We visualize samples from these models in Fig. 4 (A,B,C, respectively), confirming the findings exposed by the metrics. It is important to note that PerCo achieves the highest marginal diversity and the lowest conditional diversity; this is expected given the goal of a compression model to yield good reconstructions of the data. Obtaining high realism reconstructions allows for a good reconstruction of the real data manifold, which in turn results in high recall. However, in this case, multiple reconstructions of the same image will all look very similar, hence producing low conditional diversity.

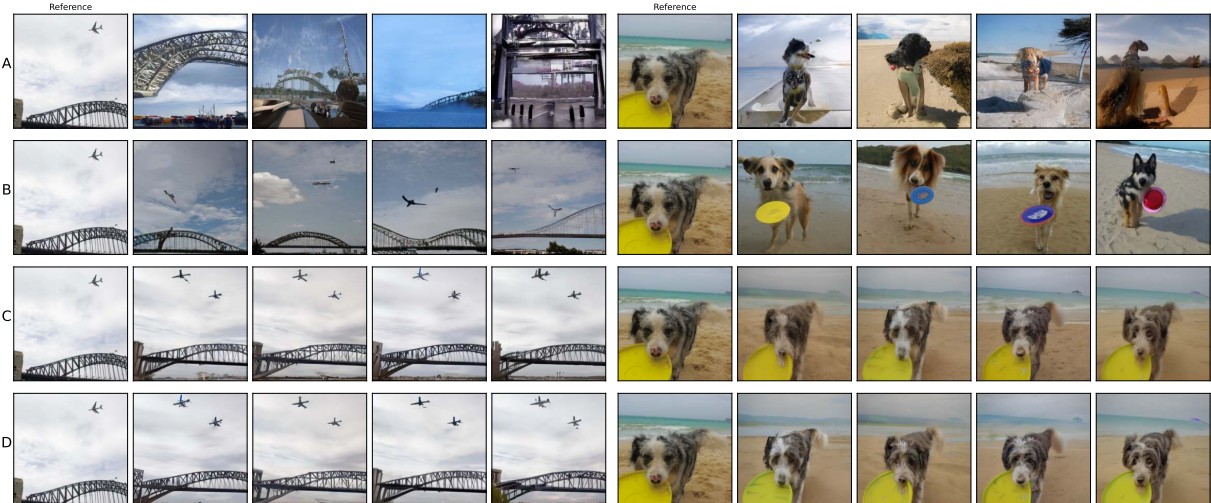

Figure 4: I-T2I qualitative results on MSCOCO2014. A-D refer to the models marked in Fig. 3. "Reference" column shows the conditioning image.

**Realism-diversity.** Considering marginal metrics in Fig. 3 (middle, top), PerCo is again the only model producing Pareto optimal points, with even higher realism (precision) and diversity (recall) scores. Also, the non-Pareto optimal points are mostly disposed along the main diagonal of the plot, suggesting rather small realism-diversity compromises. Instead, and once again similarly to the consistency-diversity case, the Pareto fronts obtained from conditional diversity and realism (middle, bottom) contain all the three models considered, with RDM model producing the most conditionally diverse samples and PerCo producing the samples with the highest conditional realism. Thus, by optimizing the model towards conditional realism, the conditional diversity is being sacrificed.

**Consistency-realism.** Similarly to T2I models, in Fig. 3 (right, top and bottom) we observe a correlation between realism (marginal or conditional) and consistency. Perhaps unsurprisingly, PerCo achieves the best results in terms of both realism and consistency, and is the only model producing Pareto optimal points. Despite not making it to the Pareto, we can still compare $LDM_{2.1\text{-}UnCLIP}$ and RDM as their hyperparameter configurations constitute two easily separable clusters, with RDM achieving much lower consistency and realism than the worst hyperparameter configuration of $LDM_{2.1\text{-}UnCLIP}$. We might attribute this difference to the different dataset scale (millions vs billions) and model capacities (400M vs. 840M) of the RDM and $LDM_{2.1\text{-}UnCLIP}$, respectively.

**I-T2I vs T2I Pareto fronts.** Comparing the two Pareto fronts, the I-T2I front exhibits lower consistency and conditional diversity, while it is higher as per marginal diversity and realism (both conditional and marginal). This can be explained by the fact that the image conditioning imposes additional constraints on the generation compared to the T2I case, resulting in an image that visually resembles the real/conditioning image more closely. On the one hand, this ensures greater realism; on the other hand, it diminishes inter-sample diversity and often distances the generation from the prototypical representation of the given text prompt, thereby reducing text-image consistency.

---

**Key insights**

- Progress in I-T2I models has been driven by improvements in realism and/or consistency. State-of-the-art models often trade realism for conditional diversity, this tradeoff is not visible when considering marginal diversity.

- Marginal diversity is dominated by the models that reconstruct more faithfully the conditioning image. This is not the case for the conditional metric that is sensitive to conditional diversity that oftentimes is important in downstream applications.

- Compression models, PerCo, should be prioritized when working on downstream applications that require high realism and consistency. However, when the downstream application requires high representation diversity, RDM and $LDM_{2.1\text{-}UnCLIP}$ are preferable.

### 3.3 Pareto fronts for geographic disparities in T2I models

Figure 5: Consistency-diversity (left), realism-diversity (middle) and consistency-realism (right) Pareto fronts for T2I models on the GeoDE dataset. Consistency measures only the presence of the object in the image. Each models' configuration differ solely for guidance scale value.

We extend the use of consistency-diversity-realism Pareto fronts to characterize potential geographic disparities of state-of-the-art conditional image generative models. In particular, we follow Hall et al. (2024) and investigate geographic disparities of T2I models using the GeoDE dataset (Ramaswamy et al., 2024).

**Consistency-diversity.** Fig. 5 (left) depicts the region-wise consistency-diversity Pareto fronts. We observe that Europe, the Americas, and Southeast Asia exhibit the best Pareto fronts, with consistently higher diversity and consistency than Africa and West Asia. As previously noted, improving diversity (computed as marginal or conditional) comes at the expense of consistency. When considering marginal metrics (top), we observe that Europe and the Americas present the best Pareto fronts. Remarkably, $LDM_{1.5}$ appears in all region-wise Pareto fronts, whereas $LDM_{2.1}$ appears remarkably less frequently, and

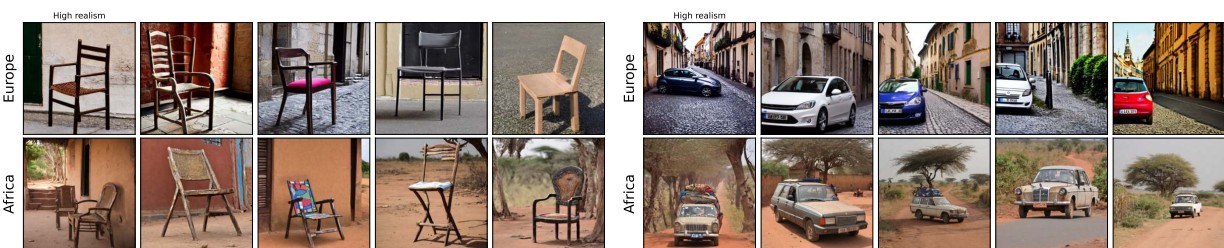

Figure 6: GeoDE qualitative. Left: `A chair in {region}`. Right: `A car in {region}`

does not appear at all in the Pareto front of Europe. This is in line with prior works that demonstrate that recent advancements on standard benchmarks may have come at the cost of reduced real world geographic representations (Hall et al., 2024). However, we positively discover that disparity *reduction* occurs via $LDM_{XL}$ which appears in the Pareto fronts of Africa, West Asia and South East Asia, bringing the results of Africa closer to those of Europe or the Americas. Yet, $LDM_{XL-Turbo}$ only appears in the Pareto fronts of some regions, and presents the highest consistency. We observe that the improvements achieved by $LDM_{XL}$ for Africa are notably reduced when distilling the model into $LDM_{XL-Turbo}$. When considering conditional metrics (bottom), we see that all T2I models appear in the Pareto fronts. Once again, $LDM_{1.5}$ shows the highest diversity and $LDM_{XL-Turbo}$ the highest consistency. As in the previous case, $LDM_{XL}$ only appears in the Pareto fronts of West Asia, Africa, and South East Asia, and bridges the consistency and diversity performance gap between Africa and both Europe and the Americas. Yet, the improvements observed in $LDM_{XL}$ for Africa disappear when considering $LDM_{XL-Turbo}$.

**Realism-diversity.** Fig. 5 (middle) depicts the region-wise realism-diversity Pareto fronts. In the top panel (precision vs. recall), we observe that, similarly to MSCOCO2014 (Fig. 1), realism and diversity performance of T2I models present a clear tradeoff. Focusing on the regions, we see that the Pareto fronts of West Asia and Africa are visibly worse than the others. In terms of models, $LDM_{1.5}$ is the model that generally dominates the Pareto fronts of all regions. Moving to conditional metrics (bottom), we notice similar trends. However, $LDM_{XL}$ appears in the highest realism part of the Pareto front of Africa, and $LDM_{XL-Turbo}$ appears in the highest realism part of the Pareto fronts of Europe and Southeast Asia. By looking at the inter-region disparities along different areas of the Pareto fronts, we notice a gradual increase of the inter-region variance when moving from high diversity (low realism) to high realism (low diversity). This result suggest that maximizing realism might exacerbate stereotypes – as suggested by the lower diversity – and increase geographical disparities – as suggested by the increased variance across region-wise Pareto fronts. We provide a visual validation of this phenomenon in Fig. 6 (See Figs. 18 and 19 in Appendix C for more examples).

**Consistency-realism.** Fig. 5 (right) depics the region-wise consistency-realism Pareto fronts. As shown in the figure, consistency and realism correlate as previously noticed on MSCOCO2014. The region-wise stratification shows that West Asia and Africa are again the regions with the worst Pareto fronts. The regions that exhibit the best Pareto fronts are East Asia, Southeast Asia, and Europe. Focusing on the top plot (marginal metrics), the Pareto fronts of all regions except the Americas contain $LDM_{1.5}$ and $LDM_{XL-Turbo}$. Note that $LDM_{1.5}$ consistently stands out in terms of realism, whereas $LDM_{XL-Turbo}$ shines in consistency. $LDM_{2.1}$ and $LDM_{XL}$ are only present in the Pareto of the Americas and Africa, respectively. In the bottom plot (conditional metrics), the situation is very similar, but we notice that for Europe and Southeast Asia the Pareto is only composed by $LDM_{XL-Turbo}$.

---

**Key insights**

- Improving generation diversity comes at the expense of consistency for all regions considered. Realism and diversity also present a clear tradeoff for all regions, whereas realism and consistency appear correlated.

- Interestingly, the oldest model, $LDM_{1.5}$ dominates the most recent ones, and consistently appears in the Pareto fronts of all regions, when considering any pair-wise objective. However, $LDM_{XL}$ reduces the disparities between Africa and Europe or the Americas in terms of diversity and consistency, as we move towards the high consistency part of the Pareto fronts.

- Advances in T2I models reduce region-wise disparities in terms of consistency and increase the disparities in terms of realism, while sacrificing diversity across all regions.

---

### 3.4   The impact of knobs on consistency-diversity-realism

Finally, in this section, we study the effect of different knobs that control consistency, diversity and realism of conditional image generative models. In the interest of space, we focus on the conditional metrics, and perform the analysis on MSCOCO2014.

**Guidance scale.** Fig. 7 depicts the effect of guidance scale on consistency-diversity (left panel), realism-diversity (middle panel), and consistency-realism (right panel) objectives. By looking at the consistency-diversity plot, we observe that increasing the guidance scale leads to improved consistency

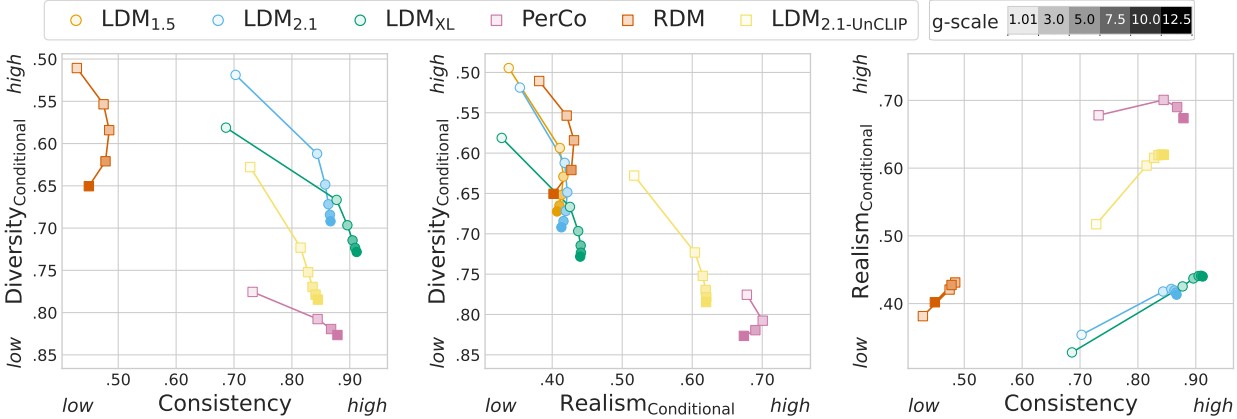

Figure 7: Ablation on guidance scale. To help readability, we report only a subset of the points presented in Figs. 1 and 3, selecting runs with default values for other knobs.

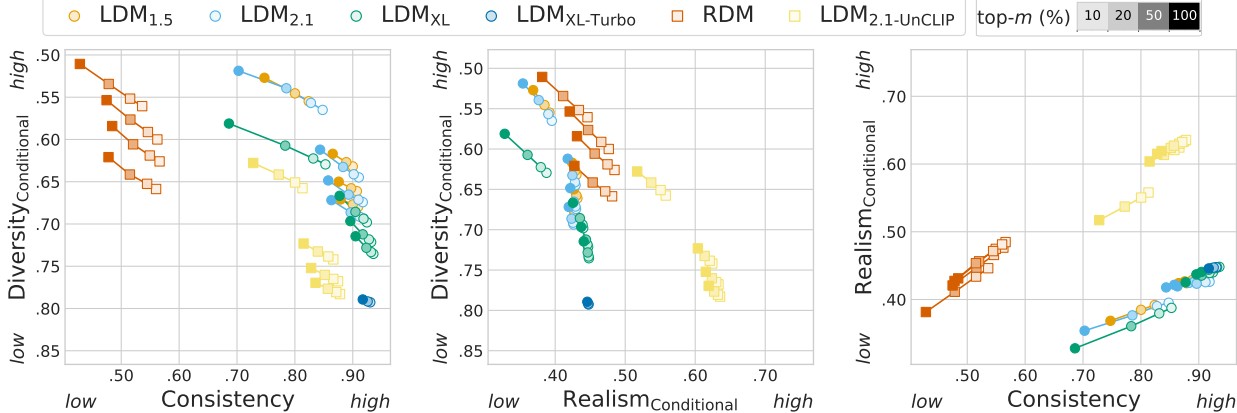

Figure 8: Ablation on top-$m$ filtering.

at the expense of the diversity in most cases [3], with $LDM_{XL}$ showing the highest relative improvements. Moreover, for all models we notice that the initial increase in the guidance scale – from 1.01 to 3.0 – leads to the biggest consistency improvements. By looking at the realism-diversity plot, we note that the increase in the guidance scale often leads to increase in realism at the expense of diversity, with $LDM_{2.1\text{-}UnCLIP}$ and PerCo benefiting the most and the least from this knob, respectively. Moreover, we note that, in most cases, increasing the guidance scales beyond 7.5 no longer results in realism improvements. Finally, the consistency-realism plot reveals that by increasing the guidance scale the models generally improve both the consistency and realism. However, too large values of guidance may lead to decreasing the image realism; this happens for all models except of $LDM_{2.1\text{-}UnCLIP}$ and $LDM_{XL}$.

**Post-hoc filtering.** Fig. 8 depicts the effect of applying top-$m$ filtering. In the consistency-diversity plot (left), we observe that top-$m$ filtering (based on CLIPScore) leads to improvements in consistency for all models – the lower the value of $m$, the higher the consistency. Unsurprisingly, the models that initially have high consistency scores do not gain as much when leveraging top-$m$ filtering as the models that start with low consistency scores. Moreover, we observe that the post-hoc filtering consistently leads to a diversity decrease. However, this decrease is less pronounced for the top-$m$ filtering than for the guidance knob, as is the case for the consistency increase (*cf*. Fig. 7). The diversity-realism plot (middle) shows that post-hoc image filtering leads to an increase in the realism at the expense of diversity. By looking at the realism-consistency plot (right), we note that the post-hoc filtering is an effective way to increase both image realism and consistency, with the latter one improving faster.

---

[3]

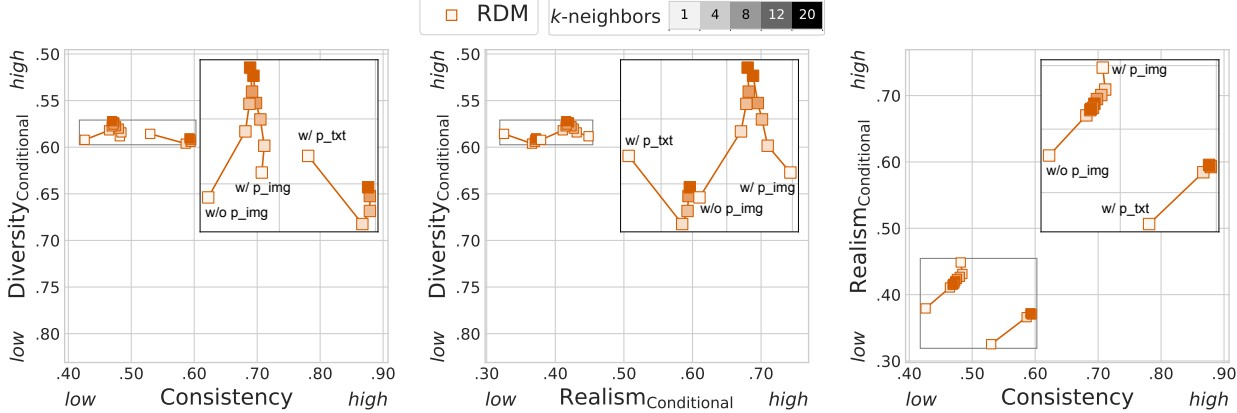

Figure 9: The effect of the neighborhood size on diversity, consistency and realism metrics. To improve readability we report a zoomed-in view in the top right of each plot.

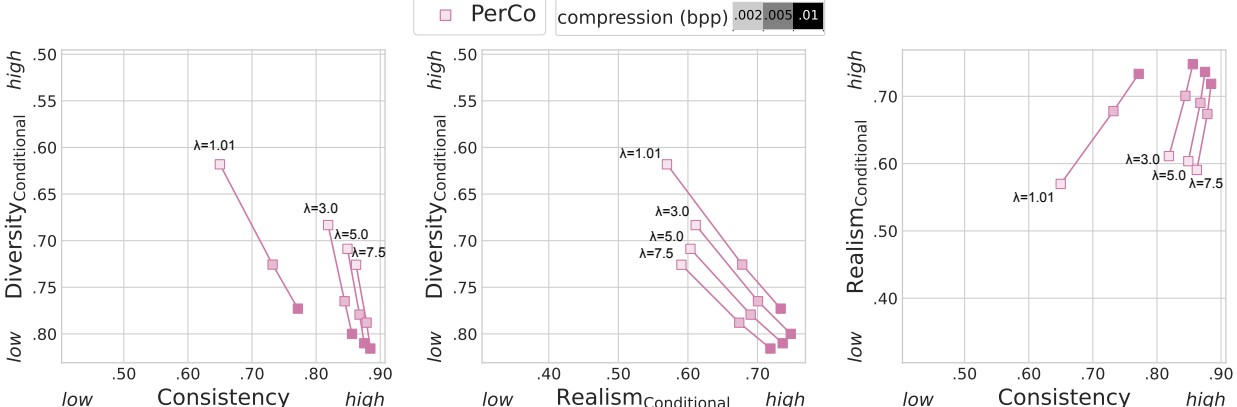

Figure 10: The effect of the compression rate on diversity, consistency and realism metrics.

**Retrieval augmentation neighborhood size.** The amount of neighbors used in retrieval augmentation may impact consistency, diversity, realism based on the semantic of the neighbors. In Fig. 9, we study the impact of the neighborhood size $k$ for RDM. We notice that, in absolute terms, the impact of $k$ is minor in all the pairs of metrics considered, suggesting that this knob is not as effective as the previous ones. In the consistency-diversity plot (left), we observe that increasing $k$ from 4 to 20 leads to a small but consistent increase in diversity, while maintaining consistency. However, when increasing $k$ from 1 to 4, we generally see a small improvement in consistency. This result is expected as by increasing the neighborhood size we might include more diverse neighbors, and as long as those neighbors are semantically similar to the query image, they will not affect the consistency of the generation. In the realism-diversity plot (middle), we observe similar trends: increasing $k$ from 4 to 20 results in small diversity improvements with little to no effect on realism, while increasing $k$ from 1 to 4 results in small realism improvements. Interestingly, RDM prompted with text achieves lower realism than the others models. Moreover, increasing $k$ when the query image is present together with the neighbors, slightly harms the realism. Finally, in the consistency-realism plot (right), we note a positive correlation between the two metrics when text query or no query is used.

**Compression rate.** The reconstructions produced by an image compression model are highly dependent on the selected compression rate, measured in terms of bit-per-pixel (bbp) of the compressed image, where high compression rate means low bpp. In Fig. 10 we assess PerCo with different bitrates and at different guidance scales. By looking at the left panel, we observe that decreasing the bitrate leads to notable increases in conditional diversity, which is inline with qualitative observations made by Careil et al. (2024). Moreover, these diversity increases only marginally reduce consistency, especially for guidance scales $> 3$, suggesting that even at high compression rates, the reconstructed images maintain their semantics. By contrast, in

realism-diversity (middle), higher compression leads to a pronounced loss in realism, suggesting that the reconstructed images do not necessarily capture all the details from the original images. Finally, the results presented for consistency-realism (right) suggest, once again, that consistency and realism are correlated.

---

**Key insights**

- Guidance scale trades diversity for consistency and realism. Consistency and realism improve with higher guidance scale, but realism improvements saturate earlier than consistency improvements.
- Post-hoc filtering improves consistency and realism at the expense of diversity. Although both consistency and realism improve with this knob, consistency increases at a faster pace. Overall, post-hoc filtering appears less effective than guidance scale.
- The effect of retrieval augmentation on consistency-diversity-realism appears minor, questioning the knobs efficacy to control the multi-objective.
- Compression rate affects image realism and diversity, but has little effect on consistency, as compression models tend to maintain the image semantics.

---

## 4    Conclusions

We proposed consistency-diversity-realism Pareto fronts as a comprehensive framework to evaluate conditional image generative models and their potential as visual world models. Using this framework, we have been able to compare several existing models on the consistency-diversity-realism axes, which allowed us to provide insights on which model is preferable over another based on the objective at hand. Our results highlighted the presence of tradeoffs among the consistency-diversity and realism-diversity axes in all the studied models. In particular, we discovered an interesting trend in the historical/temporal evolution of image generative modes, with earlier models (*e.g.*, $LDM_{1.5}$ and $LDM_{2.1}$) achieving higher diversity and more balanced tradeoffs than latest models (*e.g.* $LDM_{XL}$), which instead trade diversity to favour consistency and realism. All in all, our analysis suggested that there is no best model and the choice of model should be determined by the downstream application. We hope that Pareto fronts will become a new standard for evaluating the potential of conditional image generative models as world models.

**Limitations.** Our analysis only considers open models as evaluating closed models is very expensive or sometimes not possible. It would be interesting placing the dots of closed state-of-the-art models within the multi-objective pareto front. Moreover, it would be interesting to extend the analysis to ablate further knobs. For example, we have not included the knob of structured conditioning, like layouts, sketches or other form of control typically used to increase consistency. Another aspect that our analysis does not ablate is the effect of different data distribution on the consistency-diversity-realism pareto fronts –this aspect is currently very hard to study due to the closed data filtering recipes of most models. Furthermore, for certain evaluated knobs like the retrieval augmented generation, the analysis could be deepen by considering for example the effect of different retrieval databases or stronger/more recent models than RDM—unfortunately, there is a scarcity of open models using RAG. Finally, our work suggests future research to understand whether the observed tradeoffs are fundamental, or could be overcome by future generations of better generative models.

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

## A    Related work

The evaluation of recent state-of-the-art image generative models is often carried with human studies focusing on human preference (Ku et al., 2024; Dong et al., 2024; Kirstain et al., 2023; Otani et al., 2023; Zhou et al., 2019), where human annotators are asked to choose among images generated with different models. They are usually asked to select either the image they like the most or the image that is more aligned with the prompt used to generate it. However, due to the high cost of human annotations, works like Xu et al. (2024) use the collected human preferences to train a model to predicts them, in order to compute these metrics at lower cost. While the outcome of all these studies is useful to detect the most appealing generations, it provides only limited signal when the objective is to evaluate image generative models as world models, where several aspects need to be evaluated simultaneously. To this end, other works have focused on extending the evaluation to different aspects of the generation like fine-grained prompt-image alignment (*e.g.*, object counting and color consistency) (Ghosh et al., 2024; Hinz et al., 2020), compositionality (Li et al., 2024; Huang et al., 2023; Zhu et al., 2023; Park et al., 2021) and reasoning (Cho et al., 2023). Finally, HEIM (Lee et al., 2024) and HRS (Bakr et al., 2023) recently proposed to holistically evaluate T2I models, addressing up to 13 aspects including robustness, generalization, bias, fairness, and others, in addition to prompt-image alignment and image quality. However, some crucial aspects such as sample diversity are not investigated in these works, and more importantly, the several aspects analyzed are not combined together to understand the trade-offs and the multi-objective optimization of world models. In this regard, Yang et al. (2024); Rame et al. (2024) have investigated the multi-objective optimization in the context of finetuning foundation models including multimodal models and T2I models. In particular, these studies use Pareto fronts of multiple objectives as rewards to be directly optimized via reinforcement learning. However, none of these works considers the consistency-diversity-realism multi-objectives for conditional generative models as we do.

## B    Implementation details

Tab. 2 reports the exact knob values ablated for each model.

Table 2: Knob values ablated per model.

| Knob | values |
|---|---|
| g-scale | All LDM models: $[1.01, 3.0, 5.0, 7.5, 10.0, 12.5]$; RDM: $[1.01, 1.5, 2.0, 3.0, 5.0]$ PerCo: $[1.01, 3.0, 5.0, 7.5]$ |
| top-$m$ filtering | All but PerCo: $[10, 20, 50, 100]\%$ |
| $k$-neighbors | RDM: $[1, 4, 8, 12, 20]$ |
| comp. rate | PerCo: $[0.01, 0.005, 0.002]$bpp |

## C    Additional results

### C.1    Additional T2I results on MSCOCO2014

**Additional qualitative.** Figs. 11 to 14 depict images generated with models present in the Pareto fronts at different locations. Four models are chosen in order to provide exemplars of different areas of the Pareto: one model has the highest diversity, one has balanced consistency-diversity or realism-diversity, one has the highest consistency, and one has the highest realism. The visual comparison of the different models (different rows) validates a noticeable difference among the models in terms of consistency, diversity, and realism. Moreover, we notice that in the case of highest diversity the models tend to generate noisy images, sometimes hardly relatable with the prompt.

**Additional metrics.** Figs. 15 to 17 ablate alternatives metrics for the consistency, diversity, and realism axes. We observe no major difference with respect to the Pareto fronts reported in the main paper.

## C.2 Additional results on GeoDE

**Additional qualitative.** Figs. 18 and 19 depict images generated with models present in the Pareto fronts at different locations. Four models are chosen in order to provide exemplars of different areas of the Pareto: one model has the highest diversity, one has balanced consistency-diversity or realism-diversity, one has the highest consistency, and one has the highest realism. By comparing the different block of rows, we notice that, as consistency and realism are increased, stereotypical generations of each region get exacerbated.

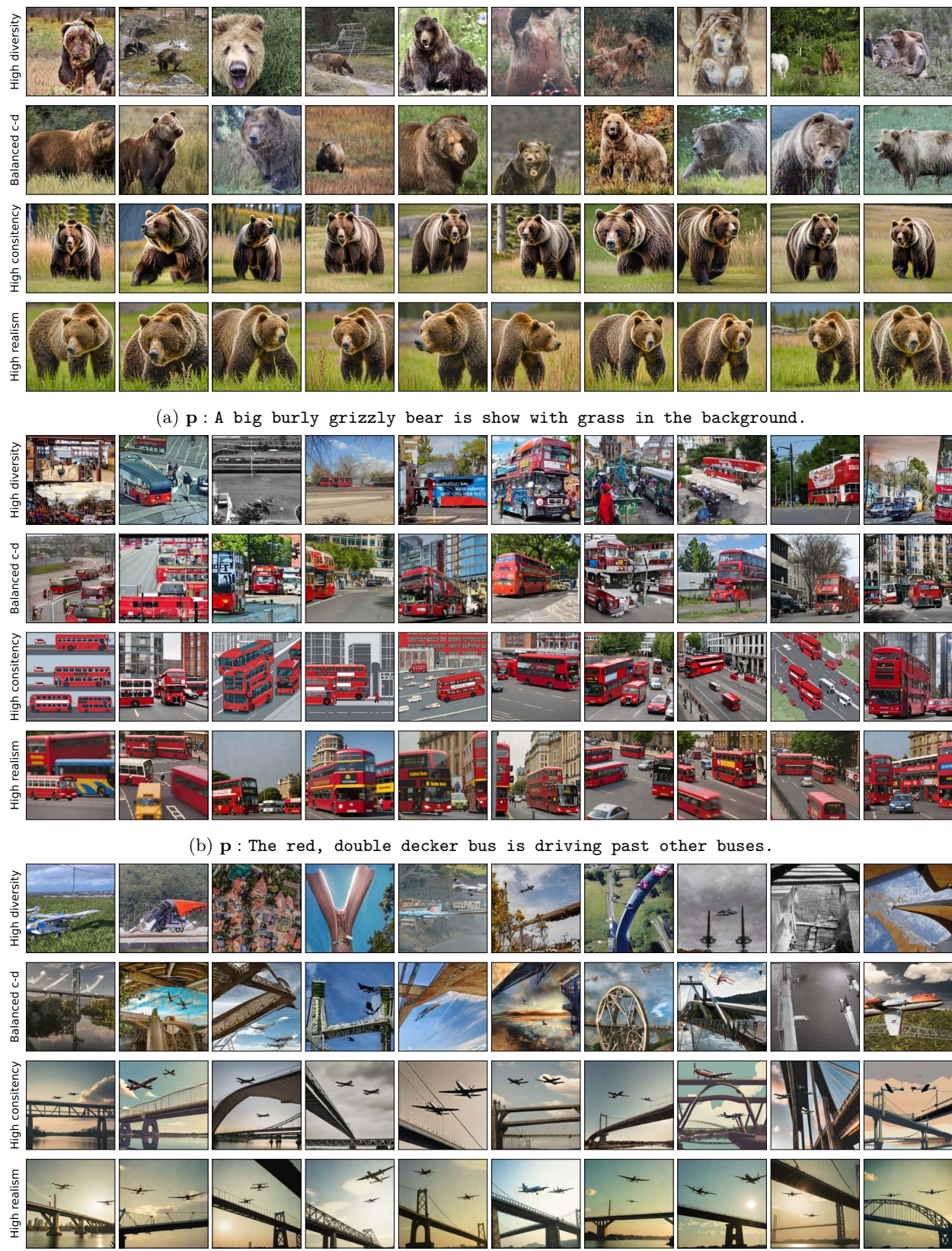

(a) **p** : `A big burly grizzly bear is show with grass in the background.`

(b) **p** : `The red, double decker bus is driving past other buses.`

(c) **p** : `Two planes flying in the sky over a bridge.`

Figure 11: "High diversity": LDM$_{1.5}$; "Balanced c-d": LDM$_{2.1}$; "High consistency": LDM$_{XL}$, "High consistency": LDM$_{XL\text{-}Turbo}$

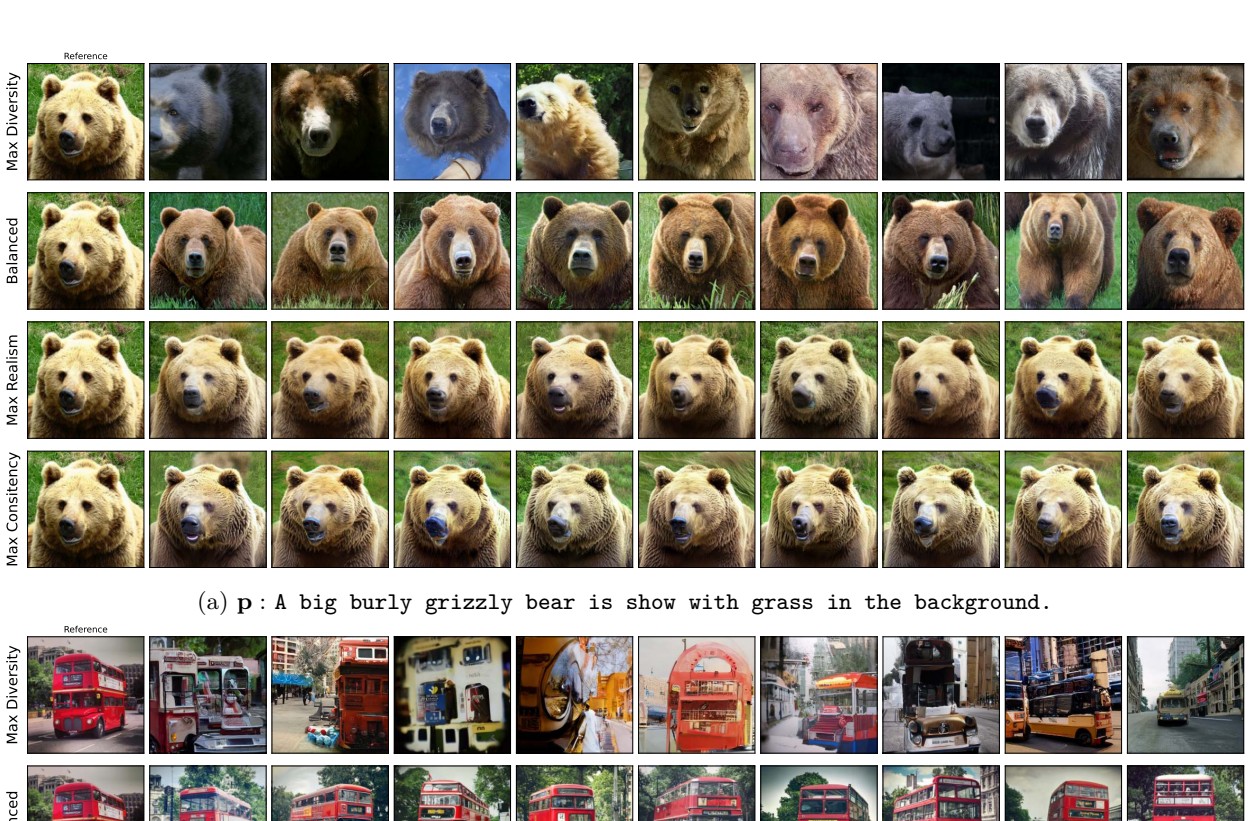

(a) **p** : A big burly grizzly bear is show with grass in the background.

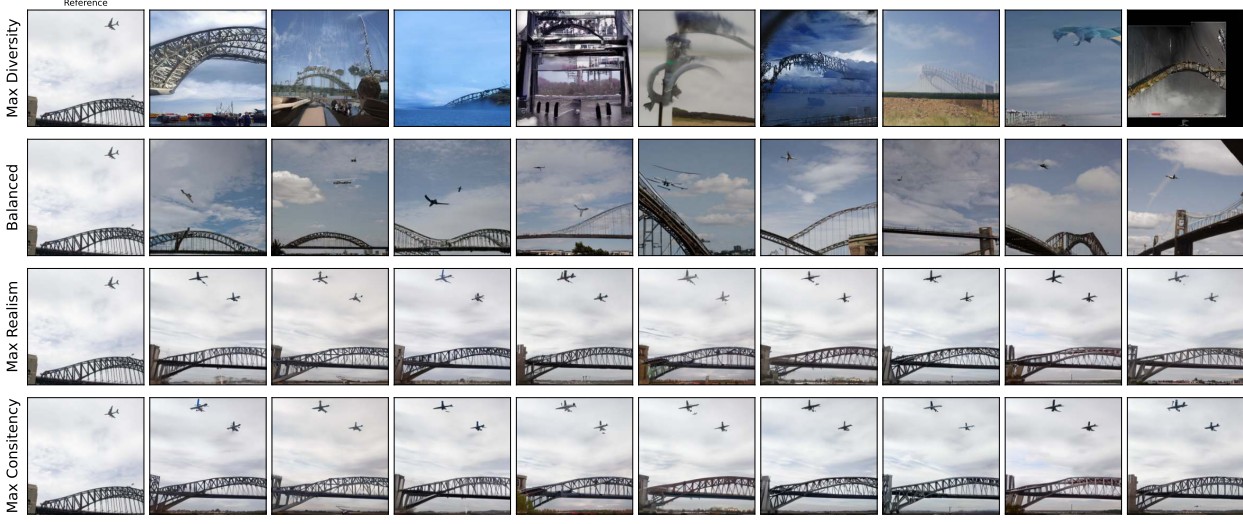

(b) **p** : The red, double decker bus is driving past other buses.

(c) **p** : Two planes flying in the sky over a bridge.

Figure 12: "High Diversity": RDM; "Balanced c-d": LDM$_{2.1\text{-UnCLIP}}$; "High consistency and realism": PerCo

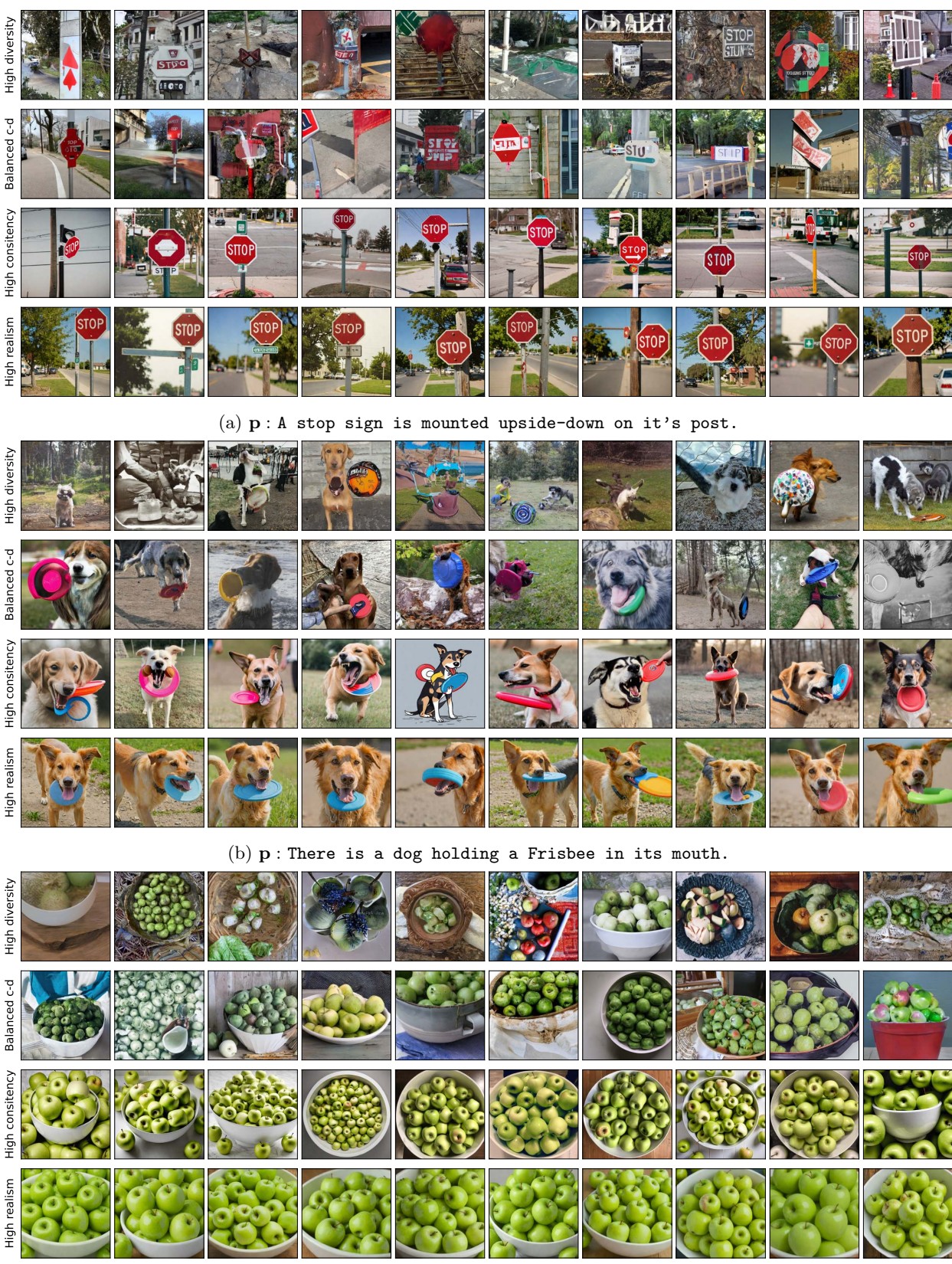

(a) **p** : `A stop sign is mounted upside-down on it's post.`

(b) **p** : `There is a dog holding a Frisbee in its mouth.`

(c) **p** : `A large white bowl of many green apples.`

Figure 13: "High diversity": LDM$_{1.5}$; "Balanced c-d": LDM$_{2.1}$; "High consistency": LDM$_{XL}$, "High consistency": LDM$_{XL\text{-}Turbo}$

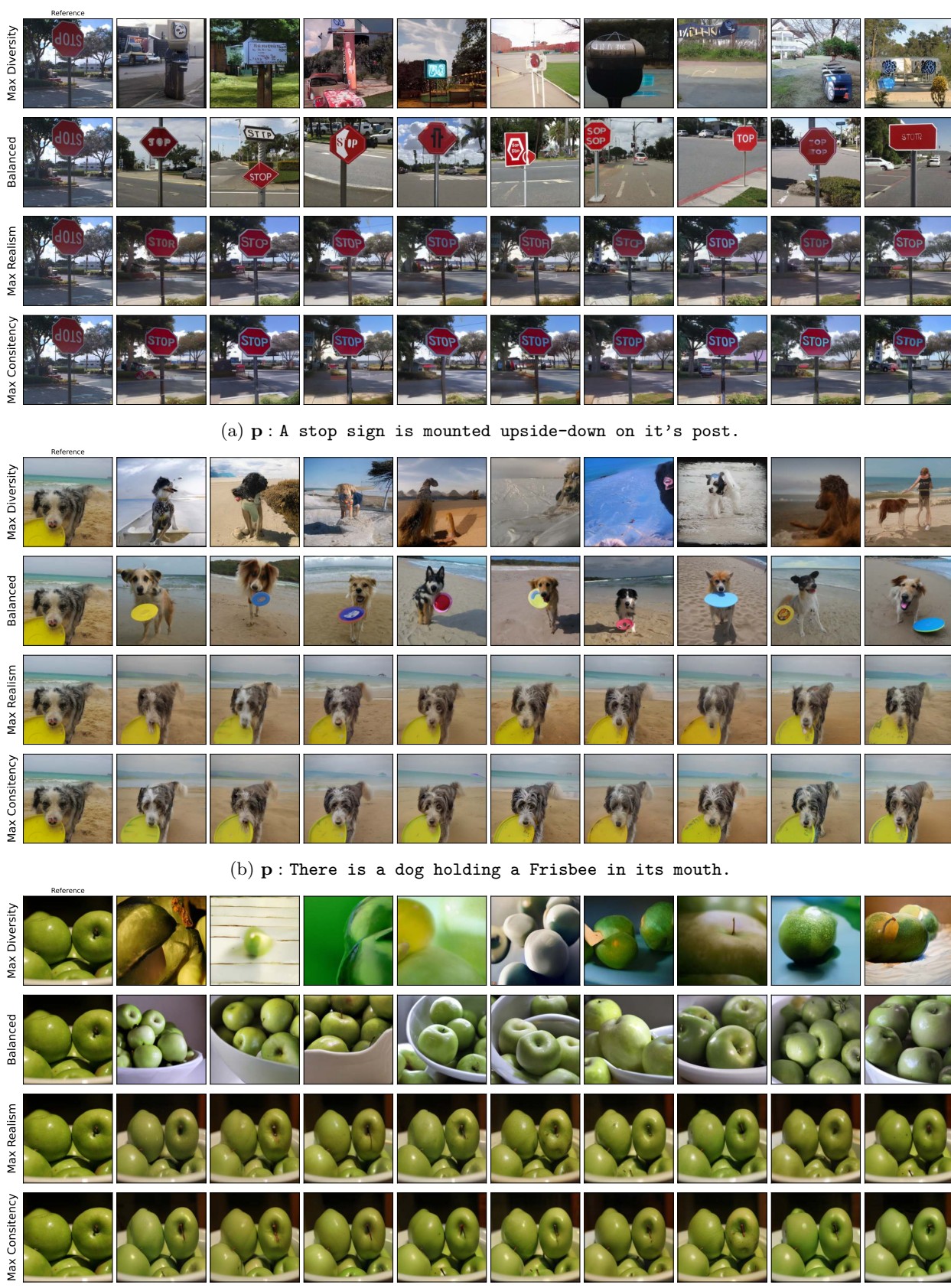

(a) **p** : A stop sign is mounted upside-down on it's post.

(b) **p** : There is a dog holding a Frisbee in its mouth.

(c) **p** : A large white bowl of many green apples.

Figure 14: "High diversity": RDM; "Balanced c-d": LDM$_{2.1\text{-UnCLIP}}$; "High consistency and realism": PerCo

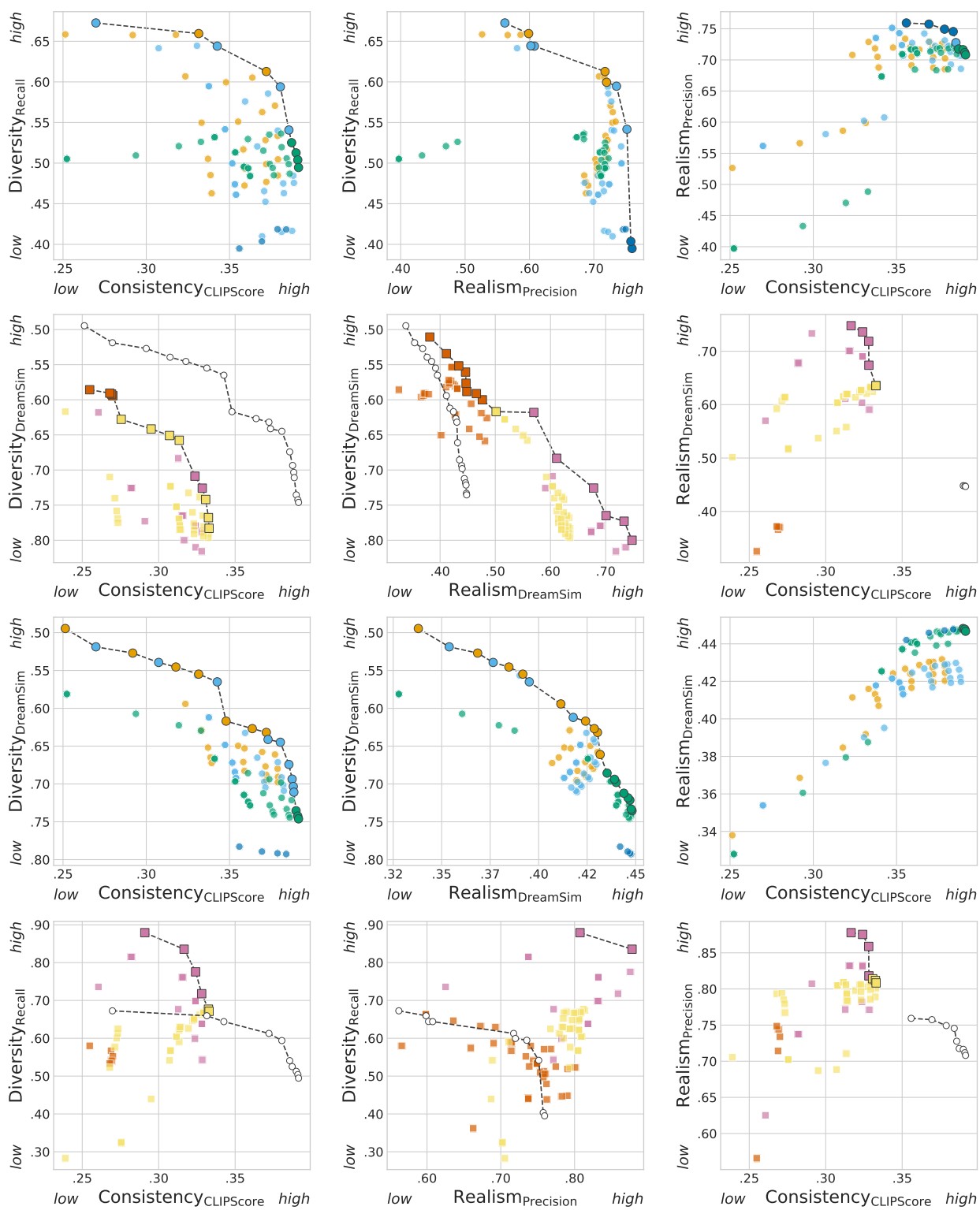

Figure 15: Using CLIPScore for consistency.

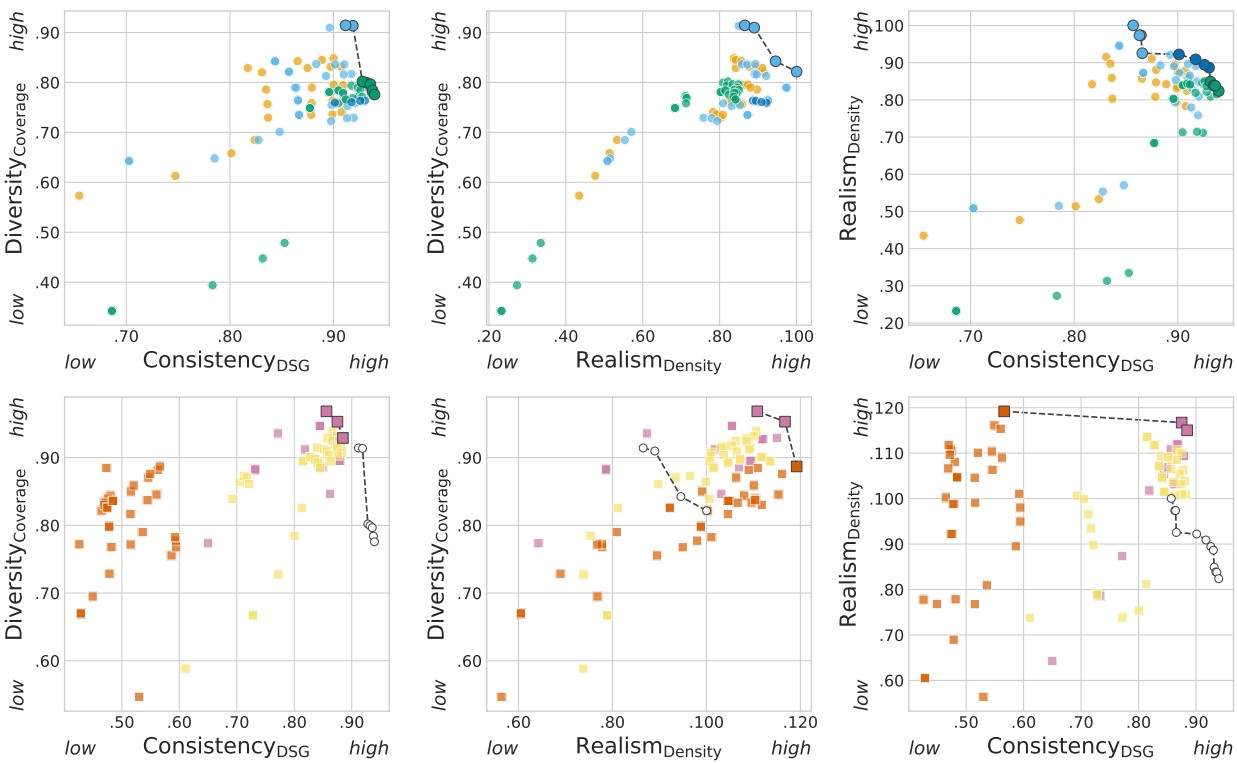

Figure 16: Using density and coverage ([Naeem et al., 2020](#)) for the marginal realism and diversity, respectively.

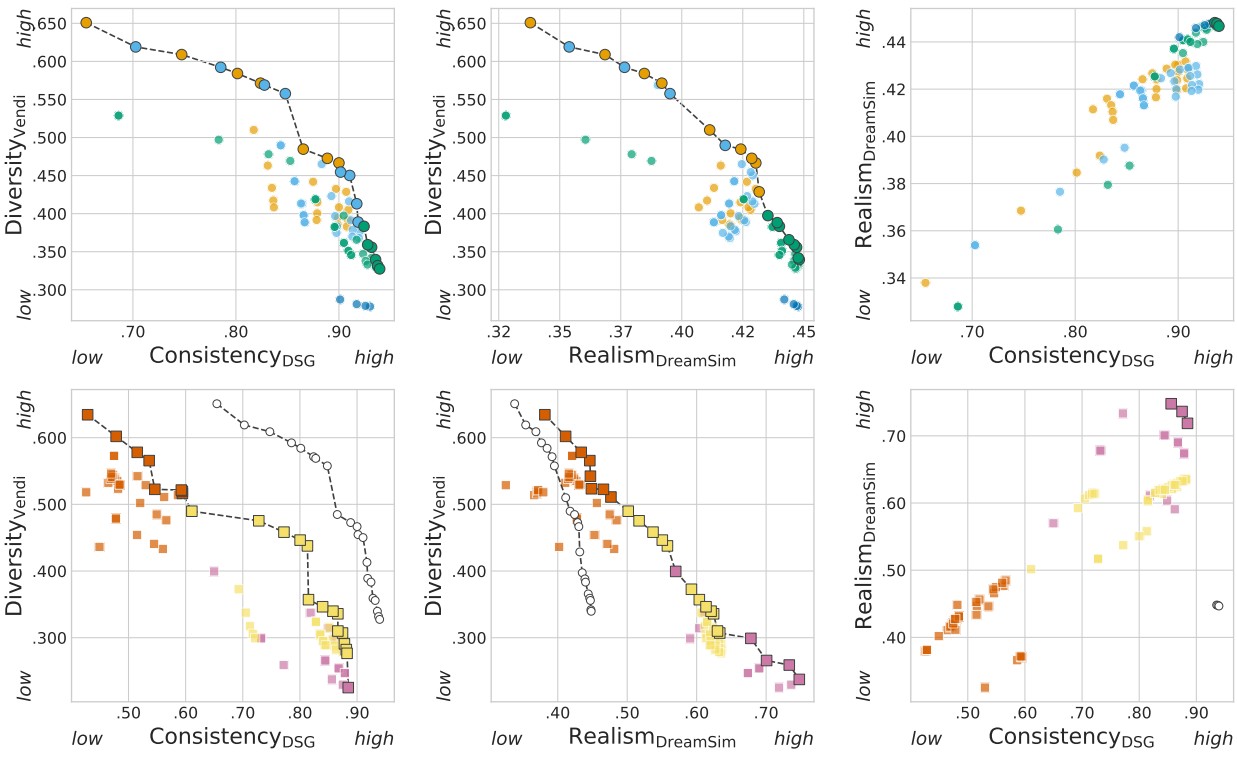

Figure 17: Using Vendi score ([Friedman & Dieng, 2023](#)) for diversity.

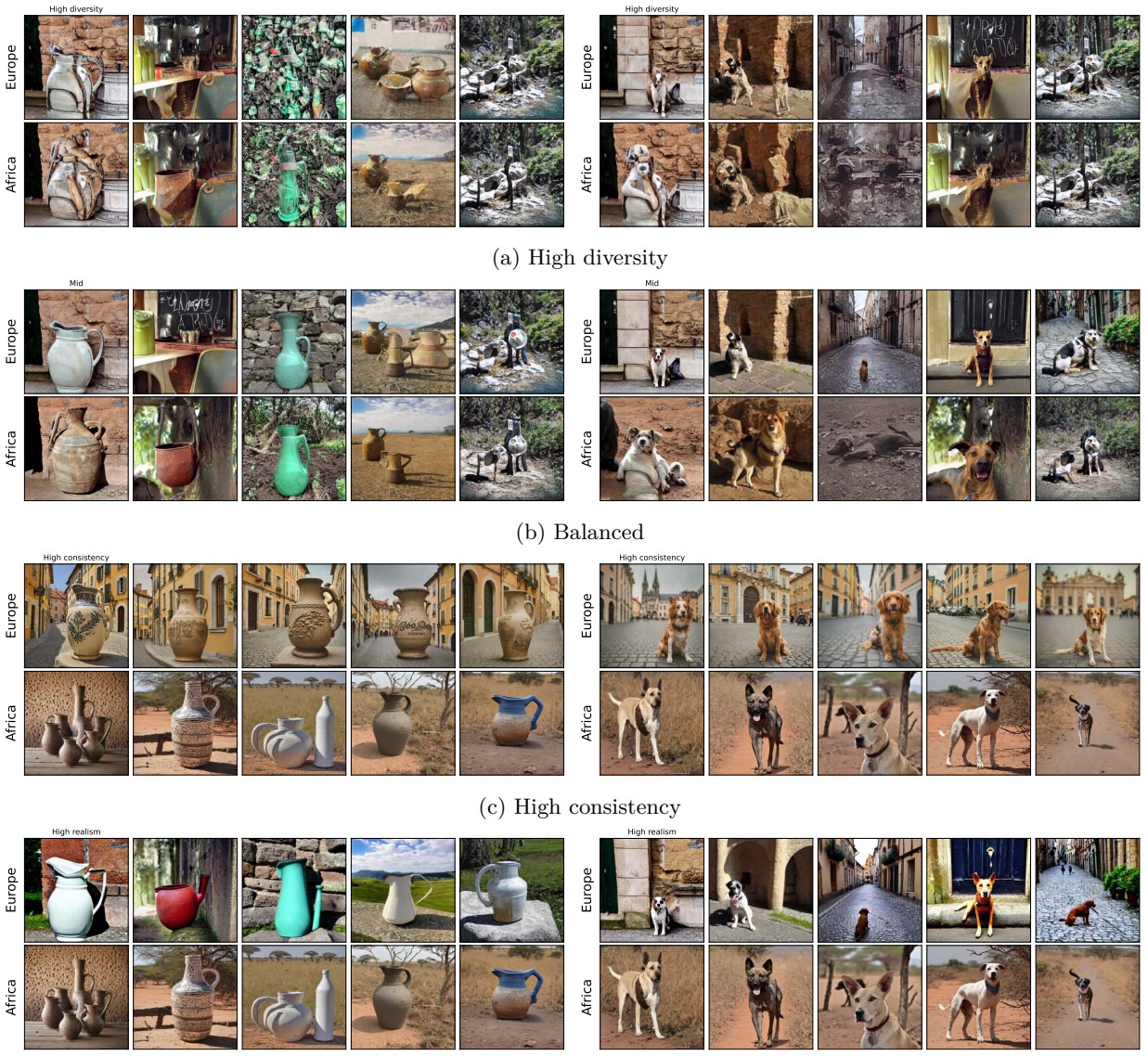

(a) High diversity

(b) Balanced

(c) High consistency

(d) High realism

Figure 18: GeoDE qualitative. Left: `A jug in {region}`. Right: `A dog in {region}`

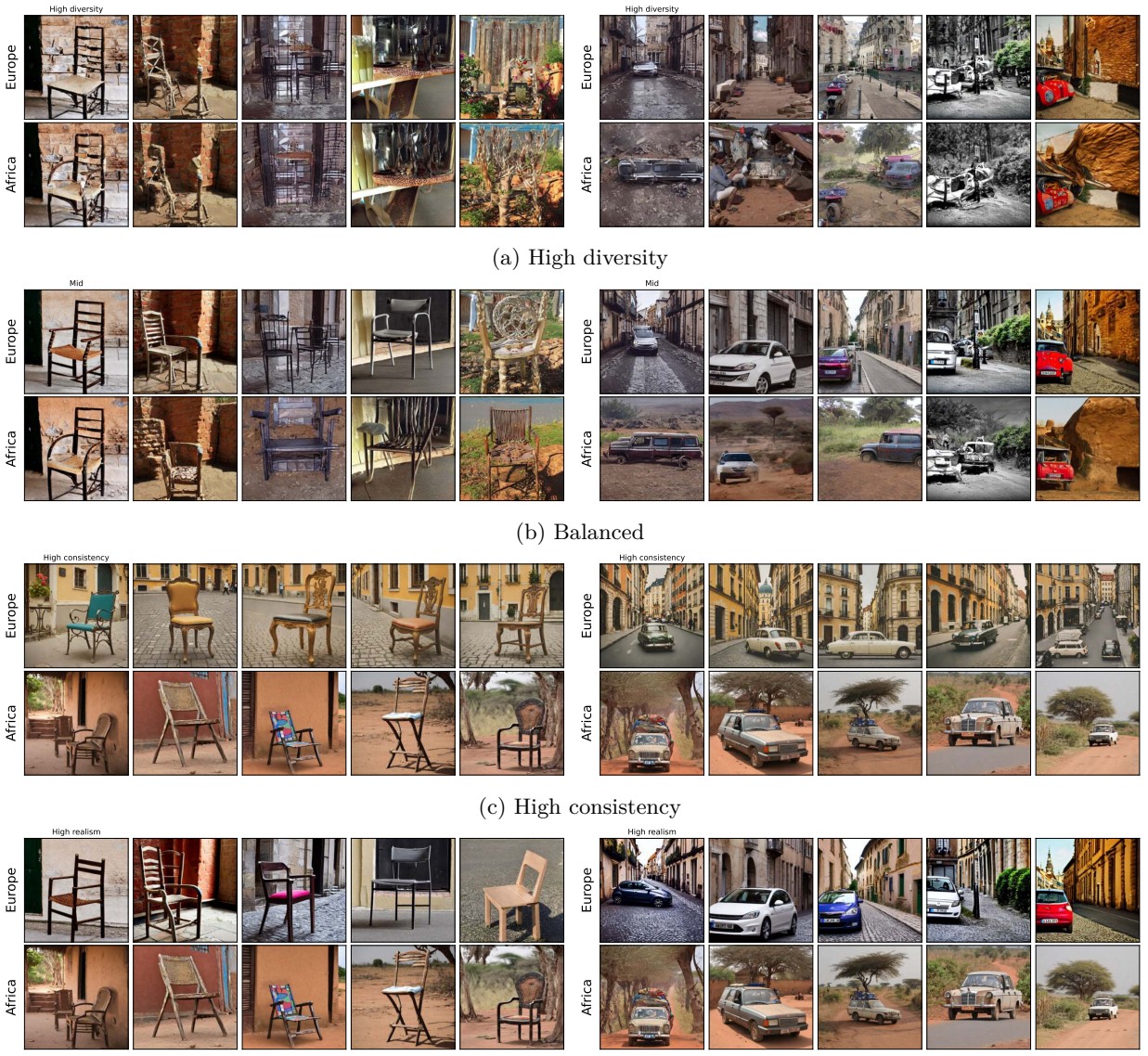

Figure 19: GeoDE qualitative. Left: `A chair in {region}`. Right: `A car in {region}`

