# OpenReview forum: "Consistency-diversity-realism Pareto fronts of conditional image generative models"
_TMLR — Rejected by TMLR_

### Review · Reviewer_hpDo · 2024-10-01

**Summary Of Contributions:**

The paper benchmarks conditional latent diffusion models on consistency, conditional and marginal realism, and conditional and marginal diversity metrics. The authors choose several recent text-to-image and image&text-to-image diffusion models and propose to consider Pareto front of the consistency-realism-diversity as a way to evaluate these models.
Experiments show a clear trade-off between diversity and consistency / realism both across models and within individual models used in different inference time regimes (referred to as knobs)

**Audience:**

Yes

**Broader Impact Concerns:**

No broader impact concerns.

**Claims And Evidence:**

No

**Requested Changes:**

1. **The authors propose to focus one three aspect of generated images: consistency-realism-diversity. But it is not clear why these aspects are necessary and/or sufficient**:
  * Please explain, why other aspects should not be considered in the framework. E.g. [1] suggest 12 different aspects that generative models would be evaluated with. Why the proposed multi-objective is comprehensive enough?

  * Please, explain the insight that one may get from the consistency-realism curves. From the provided experiments, we observe that models with better consistency are also producing more realistic images and vice versa. Therefore, it seems that it is enough to focus on the diversity vs consistency / realism trade-off.
This is something that is already evaluated with precision-recall curves in generative models literature (e.g. see Figure 7 (b) in VQ-VAE [2] or Figure 7 in Kynkäänniemi et al., 2019). However, it has not became a standard practice in T2I and T&I2I models.

2. **Ablation study**:

The authors choose a specific way to measure each of the three aspects. There are, however, many points where a different choice could have been made (e.g. other feature extractors, distance metrics, consistency metrics). I think that a solid ablation study on how these choices affect the resulting evaluation is missing.

3. **Please, add the following clarification to the methodology section**:

- Consistency metric:
  - Explain what does DSG stand for
  - Explain how the question Q_i and corresponding answer A_i are calculated
  - Define VQA() function, what is it's input and output

- Marginal diversity, Marginal realism, Post-hoc filtering:
  - Say clearly which feature extractor is used in each case. As far as I know (Kynkäänniemi et al., 2019) use pre-trained VGG-16, conditional diversity in this paper is calculated with DreamSim feature extractor and Marginal realism section mentions Inception-v3.

- Marginal diversity, Marginal realism:
  - Please, write down the formula for both metrics. Is improved precision and recall from (Kynkäänniemi et al., 2019) used or density and coverage form (Naeem et al. 2020)? Specify the hyperparameters used (e.g. number of nearest neighbours).

4. **Conclusions and insights**

The authors tend to generalise obtained results to "image generative models". However, paper only consider several versions of latent diffusions, while there are text-to-image generative models based on masked transformers [3], GANs [4] or autoregressive model [5] as well (and even wider range of unconditional generative models). Therefore, sentences like "we discovered an interesting trend in the historical/temporal evolution of image generative modes" need be reformulated.

[1] Lee, Tony, et al. "Holistic evaluation of text-to-image models." Advances in Neural Information Processing Systems 36 (2024).

[2] Razavi, Ali, Aaron Van den Oord, and Oriol Vinyals. "Generating diverse high-fidelity images with vq-vae-2." Advances in neural information processing systems 32 (2019).

[3] Chang, Huiwen, et al. "Muse: Text-to-image generation via masked generative transformers." arXiv preprint arXiv:2301.00704 (2023).

[4] Kang, Minguk, et al. "Scaling up gans for text-to-image synthesis." Proceedings of the IEEE/CVF Conference on Computer Vision and Pattern Recognition. 2023.

[5] Yu, Lili, et al. "Scaling Autoregressive Multi-Modal Models: Pretraining and Instruction Tuning." arXiv preprint arXiv:2309.02591 (2023).

**Strengths And Weaknesses:**

**Strengths:**

* Paper addresses an important existing problem: evaluation of conditional image generative model
* The authors conducted a lot of experiments: evaluating several latent diffusion models in terms of image quality and diversity and showing the effect of several so-called "generation knobs" on these metrics.

**Weaknesses**

* The authors set a very ambitious goal of providing a comprehensive framework to evaluate conditional generative models: "We proposed consistency-diversity-realism Pareto fronts as a comprehensive framework to evaluate conditional image generative models and their potential as visual world models." However,
  * Analysis is limited to latent diffusion generative models and omit other text-to-image models (e.g. autoregressive models, GANs, etc.)
  * There is no clear motivation / evidence of why exactly three chosen aspects  (consistency-diversity-realism) should be used and other potential aspects (e.g. robustness, sampling speed) should be excluded from the evaluation framework
  * There is no ablation studies: a single metric is chosen from the previous literature for each aspect
* Methodology section is not clearly written and some detail are missing (Please, see the next section)

---

> ### Author Response · Authors · 2024-10-25
> **Response**
>
> **Why consistency, diversity, and realism are sufficient?** We answer this question in the global response.
> **C-R correlation:** By looking at these two axes the conclusion is, as shown in the paper, that they have been advanced together by scaling compute and data curation. Moreover, certain algorithmic techniques of conditional diffusion models, such as guidance scale, help both dimensions by facilitating the sampling of well modeled (realistic) images while also giving more weight to the prompt in the sampling process. However, this does not mean that these two dimensions are fundamentally correlated. There are many examples in which current models still struggle to produce either realistic images in favor of cartoonish ones, or partially miss alignment with the prompt. Usually, this happens when prompts are longer and more complex, requiring strong compositionality skills by the model. In our case, neither COCO or GeoDE present challenging prompts in this sense. GenEval, Winoground-T2I, and T2I-CompBench for example show that consistency is still not there for complex scene generations.
> **Lack of novelty in P-R trade-off:** See our global response.
> **Ablation studies:** We report ablations over different implementations of both marginal and conditional metrics in Appendix C. Specifically, for consistency we show in Fig.15 that CLIPScore highlights similar correlation with Diversity and Realism as DSG, with the drawback to be less interpretable due to the high density of values in the range \[0.25, 0.4\] despite its range to be \[0,1\] —there is an almost “constant” gap between text and image in CLIP space, which hinders getting text-image similarity \> 0.5. For marginal metrics we show in Fig. 16 that Density and Coverage show similar results to Precision and Recall.  Finally, for conditional diversity in Fig.17 we substitute DreamSim with Vendi score (Friedman & Dieng, 2023\) (computed with the same feature extractor as DreamSim), which instead of averaging pairwise similarities, it computes the rank of the similarity matrix. We also experimented using CLIPScore or LPIPS for conditional diversity and realism, but we opted for DreamSim as in its paper (Fu et al., 2023\) it was shown to capture both high level and mid level features better than CLIP or LPIPS.
> **Clarification to methodology section:** Thanks for the comments, updated the pdf with the following clarifications (highlighted in red/purple in the updated pdf):
>
> - Consistency metric:
>   - DSG stands for Davidsonian Scene Graph
>   - In VQA metrics the questions and answers are usually generated with an LLM based on the caption/prompt of the image we want to evaluate. In our simplified formulation, the VQA function (vision-language model finetuned for VQA) takes as input an image (Y) and a set of binary questions (Q), and provides a prediction for each of the questions. The predictions are compared to the ground truth answers (A) to compute an accuracy. Note that in the case of DSG a dependency graph between the questions (again obtained through an LLM) is used to fix inconsistent answers by the VQA models. We refer to the DSG paper for more information. In implementation details (Section 3\) we report the specific models used for DSG.
> - Marginal diversity, Marginal realism, Post-hoc filtering:
>   - For marginal diversity and realism we use improved Precision and Recall (Kynkäänniemi et al., 2019), with 5 nearest neighbors and Inception-V3 features. We use the same features and number of neighbors when computing Density and Coverage (Naeem et al. 2020\) in supplementary materials. The conditional diversity and realism are both computed with DreamSim. Post-hoc filtering is performed based on CLIPScore as usually done.
>
> **Missing models that are not diffusion based:** We extend our claims to conditional image generative models as we consider state-of-the-art conditional image generative models. Also, LDM\_XL-Turbo despite being diffusion-based, it is adversarially distilled, showcasing a tendency to mode collapse typical of GANs. However, we do agree it would be extremely valuable to extend the analysis to non-diffusion models. Unfortunately, among the three cited works \[3,4,5\], none of them has been open sourced for public use. The only non-diffusion models that are currently available are flow-based models, e.g., Flux or LDM\_3. We are running experiments with LDM\_3, but due to the computational cost of it, we are not able to include the results in the updated pdf. Preliminary results with LDM\_3 show improved quality and consistency, but similar (low) diversity compared to LDM\_XL. We will include the full results, once completed, for the eventual camera-ready.

---

### Review · Reviewer_WUUF · 2024-10-01

**Summary Of Contributions:**

This paper provided a comprehensive understanding of multi-dimension evaluation and how the metrics trade off with each other from an empirically perspective. This work mainly focus on open source model also as author suggested that evaluating close model are much more infeasible.

**Audience:**

Yes

**Claims And Evidence:**

Yes

**Requested Changes:**

As mentioned above in weakness, but due to it's not trivial changes it's also understandable that leave those in future work

**Strengths And Weaknesses:**

-Strength: Comprehensive list of evaluation on popular Text to Image Model and Image&Text to Image Models, and also demonstrated how they correlated with each other, together with visualization on representative knobs.

-Weakness:
1. It would be great to have some initial and lighter weight conclusion on close source model such as MidJouney, or so. Even as the author mentioned that the expensiveness, close source models are even more important as a product for general users. Some preliminary findings would be beneficial
2. It would be even better to have some theory behind the conclusion, that realism are highly correlated with consistency and both of them are reversely correlated with diversity. And demonstrate if we can have a single metrics to represent both realism and consistency. Furthermore, some guideline for future model development or evaluation metrics design.

---

> ### Author Response · Authors · 2024-10-25
> **Response**
>
> **Results with closed source models:** We agree it would be beneficial for the breadth of our analysis to include close source models, however in addition to being expensive those models cannot be properly evaluated due to the inaccessibility to control knobs, e.g., guidance scale. Moreover, these models usually employ multi-step pipelines not visible to the user, which steer the generation. As a result, the steering contribution of hidden steps could not be factored out in our plots, as we did with the analyzed knobs.
> **Adding theoretical grounding to conclusions:** We find this suggestion very interesting, but a theoretical study of the interaction of these metrics is out of the scope of the current paper, and we believe it might compose a standalone paper by itself. The reason why consistency and realism correlates is, to our intuition, not principled or fundamental. For further details we refer to the response to reviewer **hpDo**

---

### Review · Reviewer_WG2c · 2024-10-11

**Summary Of Contributions:**

This work proposes consistency-diversity-realism Pareto fronts as a comprehensive framework for evaluating conditional image generative models. The evaluation protocol consists of conditional and marginal metrics:
- Conditional metrics: Consistency is measured by DSG, conditional diversity is assessed by averaging cosine similarity with the DreamSim feature extractor, and conditional realism is quantified by the maximum discrepancy between generated and real images.
- Marginal metrics: Standard precision and recall metrics.

The main finding of this work is the identification of trade-offs between the consistency-diversity and realism-diversity axes across all the studied models. The analysis suggests that there is no single best model, and the choice of model should be determined by the specific requirements of the downstream application.

**Audience:**

No

**Broader Impact Concerns:**

There is no evidence of any negative social/ethical impact associated with this work.

**Claims And Evidence:**

Yes

**Requested Changes:**

In TMLR, acceptance should be based on the level of interest this work generates within the community. In this regard, I remain concerned that many researchers studying image generative models may not find the key insights from Sections 3.1 to 3.4 particularly compelling, as they are generally considered well-known facts. If possible, could you elaborate further on why these findings are interesting enough for the community? It would be especially valuable to highlight any findings that contradict common beliefs in the field or reveal previously unreported phenomena.

**Strengths And Weaknesses:**

**Strengths**

The manuscript is well-written, with a clear description of the main findings, experiments, and conclusion. Key insights from each experiment are well-highlighted, making the main findings easily understandable.

**Weaknesses**

One of the main findings in the experiments is the identification of a trade-off between diversity and consistency or realism (although this trade-off is less clear in I-T2I models). However, this trade-off has been extensively studied and is widely known within the community. References to this trade-off can be found in:
- Section 3.1 of the BigGAN paper (https://arxiv.org/pdf/1809.11096).
- Figure 3 and Table 8 of the classifier-free guidance paper (https://arxiv.org/pdf/2207.12598).

Additionally, this trade-off has been observed across different generative methodologies, including GANs, autoregressive models, and diffusion models. In the case of GANs, truncation parameters control this trade-off, for autoregressive models, it is the temperature, and for diffusion models, it is the guidance scale. Therefore, I am not convinced that the main findings of this work significantly advance the community’s understanding in this area.
Another weakness of this work is the lack of comparison to existing image generation evaluation frameworks. Recently, several proposals have been made to evaluate T2I models from a more holistic perspective. Among them, I find the following work particularly valuable to the community:
- Holistic Evaluation of Text-to-Image Models (NeurIPS’23 Track on Datasets and Benchmarks).

This evaluation framework systematically assesses models across 12 perspectives and has evaluated 26 state-of-the-art models, including proprietary ones. Therefore, I am concerned about the advantages of the newly proposed evaluation framework compared to this existing work.

---

> ### Author Response · Authors · 2024-10-25
> **Response**
>
> **Diversity-consistency and diversity-realism trade-off already well-known:** We refer the reviewer to the global response on this matter.
> **Lack of comparison with HEIM:** We again refer the reviewer to the main response.
> **Lack of audience:** We respectfully disagree with the reviewer. In the main response we provide multiple reasons that remark why our work is valuable and needed by the community. Moreover, we would comment that the objective of a scientific paper should not only be contradicting common belief, but in most cases it is to validate either empirically or theoretically phenomena that researchers/practitioners have perhaps qualitatively intuited. In our case, by experimenting with state-of-the-art models we noticed clear lacks in terms of diversity, despite models being able to produce images of amazing quality. Thanks to the systematic evaluation of several models and knobs using Pareto fronts we have been able to confirm an undesirable trend, where models that are trained with more parameters, on larger data distributions, and with more refined architectures and recipes, end up in trading diversity to favor realism and consistency. This is not trivial and not explainable by D-R or C well-known trade-off for certain knobs. This finding is also enriched with the geodiversity analysis, the adoption of different metrics, and the comparison with models conditioned on different modalities. The proposed Pareto fronts equips the community with a tool that clearly exposes the frontier that needs to be pushed in order to advance the state of the art in world representation with conditional image generative models.

---

### Author Response · Authors · 2024-10-25
**Global response**

We thank the reviewers for their feedback. We are glad to hear their remarks on the importance of the problem (**hpDo**), the extensiveness of the experimental results (**WUUF, hpDo**), and the clarity of the manuscript (**WG2c**).

We identified two shared concerns raised by reviewers **WG2c** and **hpDo**, which we would like to address here:

1. ***Trade-off between diversity (D) and realism (R) or consistency (C) is already well-known from previous literature, e.g., BigGAN (Brock et al., 2018), VQ-VAE (Razavi et al., 2019), CFG (Ho & Salimans, 2021\)***.  We are aware of this evidence and indeed we mention it in the introduction when presenting the knobs, e.g. ”*For example, it has been shown that the guidance scale in classifier free guidance of diffusion models (Ho & Salimans, 2021), trades image fidelity for diversity (Saharia et al., 2022; Corso et al., 2024\)*”. However, in our study, we focus on benchmarking state-of-the-art conditional image generative models from a world modeling perspective. Pareto fronts are proposed as a tool to build evidence on the potential of current models to operate as world simulators. Leveraging Pareto fronts allows us to extend current evaluations by acknowledging that conditional image generation is a multi-objective problem, and as such, there is a set of optimal solutions in the space of objective functions, which present compromises. Sweeping over multiple knobs, conditioning modalities and models provides a rich understanding on the trade-offs of the current landscape of conditional image generative models. Specifically, our analysis reveals that although more recent models improve on quality and consistency they do not outperform older models in terms of image diversity — making the newer models less suited as world models than the previously developed models. To the best of our knowledge this is a new observation and by introducing Pareto fronts we are inviting the research community to contribute Pareto optimal models that will not trade representation diversity for generation quality and consistency.
2. ***Comparison with Holistic Evaluation of text-to-Image Models (HEIM) (Lee et al., 2024\) and choice of adopting only C-D-R axes***. We compare with HEIM in Appendix A of the manuscript: “*Finally, HEIM (Lee et al., 2024\) and HRS (Bakr et al., 2023\) recently proposed to holistically evaluate T2I models, addressing up to 13 aspects including robustness, generalization, bias, fairness, and others, in addition to prompt-image alignment and image quality. However, some crucial aspects such **as sample diversity** are not investigated in these works, and more importantly, the several aspects analyzed are not combined together to understand the trade-offs and the multi-objective optimization of world models.*” We note that HEIM may propose 12 evaluation axes; yet 50% of these axes rely on CLIPScore. The tabular presentation of the 12 evaluation results hinders the ability of ML practitioners to draw conclusions on the models that would help define possible future model improvements, i.e., there is no information of multi-objective dynamics, which instead we capture via Pareto fronts. Moreover, these evaluation axes are not tailored to understanding the potential of conditional generative models as world models. In our case, the necessity and sufficiency of C-D-R axes to evaluate world models is justified in the intro: “*World models aim to represent the real world as accurately and comprehensively as possible. Therefore, visual world models should not only be able to yield high quality image generations, but also generate content that is representative of the diversity of the world, while ensuring prompt consistency.*” Although C-D-R axes may not be exhaustive, it is worth noting that many of the 12 evaluation axes presented in HEIM can be related to them, but none of 12 axes capture either marginal or conditional diversity properly. For example, “alignment, reasoning, and knowledge” in HEIM are captured by CLIPScore, highlighting their strong connection with consistency. In our case, consistency is computed with VQA-based metrics, which target those axes through questions (e.g. “Does George Washington appear in the image?”, “Is there a bird/potted-plant in the image? Is the potted-plant below the bird?”).  Then, “fairness” and “bias” are computed as the equivariance of CLIPScore, which is relatable to our investigation of geodiversity. “Quality” as measured by FID and IS partially maps to our marginal realism (Precision). Other dimensions, like originality, robustness, toxicity, or multilingualism are not captured by C-D-Q and could be taken into account as future extensions. Instead, dimensions like efficiency or aesthetics might be less relevant for evaluating a world model.

---

### Decision · Action_Editor_JR78 · 2024-12-09

**Recommendation:** Reject

**Comment:**

Among the three reviewers, two are borderline, and one is negative. The borderline reviewers (Reviewer WG2c and Reviewer WUUF did have some concerns on the reasonableness of the evaluations. And they do not seems to express strong opinions to the acceptance of the paper. The negative reviewer posted several concerns on the paper, including the ambitiousness of the paper (the paper targets at the general conditional image generative models but only evaluated on a limited number of LDM and related models), the lack of ablation study  on why the three aspects of consistency-realism-diversity is appropriated, as well as the presentation. The reviewer considered the rebuttal did not fully address his/her concerns. Specifically, the reviewer is not convinced that there is sufficient evidence in the paper to support the claim that this is a comprehensive framework that should become the new standard for evaluating conditional generative models.

I suggest the authors to revise the paper according to the reviews, especially consider adding more comprehensive experiments and motivate the paper better.

**Audience:**

Researchers in diffusion generative models is likely to find the paper to be worthwhile reading.

**Claims And Evidence:**

This paper makes several claims, for example, trade-off between diversity and consistency/realism, evaluating T2I models with consistency-realism-diversity metrics, no global best models to achieve best in all metrics. Some of the claims are found to be insufficiently supported by the evidence in the paper. For example, it is not clear why consistency-realism-diversity but not others are the right metrics for evaluate, and the models used in the evaluations are not diverse enough to draw affirmative conclusions.